# Do Membership Inference Attacks Work on Large Language Models?

Michael Duan[*1]     Anshuman Suri[*2]
Niloofar Mireshghallah[1]     Sewon Min[1]     Weijia Shi[1]     Luke Zettlemoyer[1]
Yulia Tsvetkov[1]     Yejin Choi[1]     David Evans[2]     Hannaneh Hajishirzi[1,3]
[1]University of Washington     [2]University of Virginia     [3]Allen Institute for AI
<micdun@cs.washington.edu>, <as9rw@virginia.edu>

## Abstract

Membership inference attacks (MIAs) attempt to predict whether a particular datapoint is a member of a target model's training data. Despite extensive research on traditional machine learning models, there has been limited work studying MIA on the pre-training data of large language models (LLMs). We perform a large-scale evaluation of MIAs over a suite of language models (LMs) trained on the Pile, ranging from 160M to 12B parameters. We find that MIAs barely outperform random guessing for most settings across varying LLM sizes and domains. Further analyses reveal that this poor performance can be attributed to (1) the combination of a large dataset and few training iterations, and (2) an inherently fuzzy boundary between members and non-members. We also find that, when LLMs have been shown to be vulnerable to MIAs, this apparent success can be attributed to a distribution shift, e.g., members and non-members are seemingly drawn from identical domain but with different temporal ranges. Finally, we observe that existing MIAs are highly sensitive to even small changes in a sample. Such changes may cause samples that are lexically or semantically similar to members to be classified as non-members, which may be at odds with leakage that privacy auditors care about. We release our code and data as a unified benchmark package that includes all existing MIAs, supporting future work.

## 1   Introduction

Membership inference attacks (MIAs) aim to predict whether a particular record belongs to the training dataset of a given model. Thus, MIAs have great utility for privacy auditing of models (Steinke et al., 2023), as well as investigating memorization of training data, copyright violations and test-set contamination (Shi et al., 2023; Oren et al., 2023). While MIAs have been found to achieve high attack performance, alluding to high levels of training-data memorization (Zarifzadeh et al., 2023; Bertran et al., 2023; Lukas et al., 2023), most analyses are limited to classifiers or LM fine-tuning (Mireshghallah et al., 2022b; Fu et al., 2023). The performance of existing MIAs on LLMs and their pre-training data is largely unexplored. In this work, we set out to explore the challenges in evaluating membership inference attacks on LLMs, across an array of five commonly-used membership inference attacks. We introduce MIMIR[1], a unified repository for evaluating MIAs for LMs, with implementations of several attacks from literature. We report on experiments extensively evaluating these MIAs against target models from the Pythia suite (Biderman et al., 2023b) over the Pile (Gao et al., 2020) (§3). For the most part, we find that the performance across most MIAs and target domains is *near-random*.

Our further analysis suggests that the inherent characteristics of LLMs at scale—specifically, the use of massive training data and near-one epoch training (§3.2.1)—considerably decrease

---

[*]Equal Contribution.
[1]http://github.com/iamgroot42/mimir

current MIA performance. This suggests that the success of current MIAs in previous settings does not transfer well to attacking pre-trained LLMs seemingly due to a lack of memorization of member data. We also find that the frequent overlap between members and non-members from natural language domains considerably decreases MIA performance and raises the question of how membership should be interpreted (§3.2.2). Notably, in several domains, non-members have high $n$-gram overlap with members, e.g., non-members from the Pile Wikipedia and ArXiv test samples have average 7-gram overlaps of over 30%. Notably, non-members with lower $n$-gram overlap are more distinguishable by existing MIAs. We also suggest that high MIA performance reported by prior work (Shi et al., 2023) is likely because non-members are chosen from the same domain as members but are temporally shifted, and these seemingly in-domain non-members likely belong to a different distribution as a result of $n$-gram overlap shift (§4).

Finally, building off membership ambiguity due to $n$-gram overlap, we discuss how the precise definition of members in standard MI may not capture important information leakage under generative text-modeling. We generate modified members preserving lexical and/or semantic similarity by altering a tiny fraction of tokens and show that existing MIAs classify them as non-members with a high degree of confidence, often more definitively than actual non-members (§5). We encourage future work to study MI using membership definitions accounting for such fuzzy members to better understand privacy leakage.

## 2 Background

The goal of an MIA is to infer whether a given data point $x$ was part of the training dataset $\mathcal{D}$ for model $\mathcal{M}$, by computing a membership score $f(\mathbf{x}; \mathcal{M})$. This score is then thresholded to determine a target sample's membership.

MIAs are often used as a proxy to determine whether a machine-learning model leaks information related to its training data (Shokri et al., 2017; Shokri, 2022; Cummings et al., 2024). It is the de-facto threat model when discussing machine-learning privacy (Shokri et al., 2017), with a large array of attacks (Yeom et al., 2018; Carlini et al., 2022; Mireshghallah et al., 2022a) and defenses (Abadi et al., 2016; Tang et al., 2022; Chen et al., 2022). More involved approaches include training shadow models (Shokri et al., 2017; Ye et al., 2022) on non-overlapping data from the target model's underlying data distribution. While attacks like LiRA (Carlini et al., 2022) show promise, they require training multiple copies of shadow models, which is often intractable for LLMs. Other stronger assumptions for MIAs include white-box access to the model (i.e., access to model parameters) or access to a ground-truth subset of member/in-distribution non-member samples for training meta-classifiers.

In our setting, $\mathcal{M}$ is an auto-regressive language model that outputs a probability distribution of the next token given a prefix, denoted as $P(x_t | x_1...x_{t-1}; \mathcal{M})$. We consider five MIAs (See Appendix A.4 for detailed descriptions):

(1) **LOSS** (Yeom et al., 2018) - the target sample's loss under the model: $f(\mathbf{x}; \mathcal{M}) = \mathcal{L}(\mathbf{x}; \mathcal{M})$.

(2) **Reference-based** (Carlini et al., 2021) calibrates $\mathcal{L}(\mathbf{x}; \mathcal{M})$ with respect to another *reference model* ($\mathcal{M}_{ref}$) to account for the intrinsic complexity of the target sample $\mathbf{x}$: $f(\mathbf{x}; \mathcal{M}) = \mathcal{L}(\mathbf{x}; \mathcal{M}) - \mathcal{L}(\mathbf{x}; \mathcal{M}_{\text{ref}})$.

(3) **Zlib Entropy** (Carlini et al., 2021) calibrates $\mathcal{L}(\mathbf{x}; \mathcal{M})$ with target sample $\mathbf{x}$'s zlib compression size: $f(\mathbf{x}; \mathcal{M}) = \mathcal{L}(\mathbf{x}; \mathcal{M})/\text{zlib}(\mathbf{x})$.

(4) **Neighborhood attack** (Mattern et al., 2023) - the curvature of the loss function at $\mathbf{x}$, estimated by perturbing the target sequence to create $n$ 'neighboring' samples, and comparing the loss of the target $\mathbf{x}$ with its neighbors $\tilde{\mathbf{x}}$: $f(\mathbf{x}; \mathcal{M}) = \mathcal{L}(\mathbf{x}; \mathcal{M}) - \frac{1}{n}\sum_{i=1}^{n}\mathcal{L}(\tilde{\mathbf{x}}_i; \mathcal{M})$.

(5) **Min-$k$% Prob** (Shi et al., 2023) uses the $k\%$ of tokens with the lowest likelihoods to compute a score instead of averaging over all token probabilities as with LOSS: $f(\mathbf{x}; \mathcal{M}) = \frac{1}{|\text{min-}k(\mathbf{x})|}\sum_{x_i \in \text{min-}k(\mathbf{x})} -\log(p(x_i \mid x_1, ..., x_{i-1}))$.

| # Params | Wikipedia | | | | | Github | | | | | Pile CC | | | | | PubMed Central | | | | |
|---|---|---|---|---|---|---|---|---|---|---|---|---|---|---|---|---|---|---|---|---|
| | LOSS | Ref | min-$k$ | zlib | Ne | LOSS | Ref | min-$k$ | zlib | Ne | LOSS | Ref | min-$k$ | zlib | Ne | LOSS | Ref | min-$k$ | zlib | Ne |
| 70M | .503 | .504 | .494 | .508 | **.510** | .629 | .584 | .627 | **.648** | .635 | .494 | .489 | **.503** | .495 | .489 | .502 | **.516** | .510 | .502 | .485 |
| 160M | .504 | **.515** | .488 | .514 | .513 | .638 | .591 | .634 | **.656** | .638 | .497 | .497 | **.503** | .498 | .496 | .500 | **.516** | .504 | .500 | .486 |
| 1.4B | .510 | **.544** | .506 | .518 | .518 | .656 | .587 | .654 | **.670** | .650 | .500 | **.525** | .509 | .502 | .499 | .496 | **.530** | .505 | .500 | .490 |
| 2.8B | .516 | **.565** | .511 | .522 | .517 | .707 | .657 | .708 | **.717** | .698 | .501 | **.537** | .509 | .503 | .502 | .498 | **.536** | .502 | .500 | .497 |
| 6.9B | .514 | **.571** | .512 | .521 | .514 | .672 | .573 | .675 | **.684** | .654 | .511 | **.564** | .516 | .512 | .505 | .504 | **.552** | .508 | .504 | .497 |
| 12B | .516 | **.579** | .517 | .524 | .520 | .678 | .559 | .683 | **.690** | .660 | .516 | **.582** | .521 | .517 | .514 | .506 | **.559** | .512 | .506 | .497 |

| # Params | ArXiv | | | | | DM Math | | | | | HackerNews | | | | | The Pile | | | | |
|---|---|---|---|---|---|---|---|---|---|---|---|---|---|---|---|---|---|---|---|---|
| | LOSS | Ref | min-$k$ | zlib | Ne | LOSS | Ref | min-$k$ | zlib | Ne | LOSS | Ref | min-$k$ | zlib | Ne | LOSS | Ref | min-$k$ | zlib | Ne |
| 70M | **.506** | .481 | .499 | .495 | .496 | .492 | **.520** | .495 | .485 | .481 | .494 | .495 | **.507** | .497 | .506 | .503 | **.511** | .508 | .506 | .499 |
| 160M | **.507** | .486 | .501 | .500 | **.507** | .490 | **.523** | .493 | .482 | .489 | .492 | .490 | .497 | .497 | .499 | .502 | **.511** | .506 | .505 | .499 |
| 1.4B | **.513** | .510 | .511 | .508 | .511 | .486 | **.512** | .497 | .481 | .465 | .503 | **.514** | .509 | .502 | .504 | .504 | **.521** | .508 | .507 | .504 |
| 2.8B | .517 | **.531** | .522 | .512 | .519 | .485 | **.504** | .497 | .482 | .467 | .510 | **.549** | .518 | .507 | .513 | .507 | **.530** | .512 | .510 | .506 |
| 6.9B | .521 | **.538** | .524 | .516 | .519 | .485 | **.508** | .496 | .481 | .469 | .513 | **.546** | .528 | .508 | .512 | .510 | **.549** | .516 | .512 | .510 |
| 12B | .527 | **.555** | .530 | .521 | .519 | .485 | **.512** | .495 | .481 | .475 | .518 | **.565** | .533 | .512 | .515 | .513 | **.558** | .521 | .515 | .511 |

Table 1: AUC ROC of MIAs against PYTHIA-DEDUP (TPR@low%FPR results in Table 11). Highest performance across different MIAs is bolded per domain. **MIA methods perform near random ($< .6$) in most domains.** See Appendix B.3 for GitHub outlier discussion.

**Membership Inference vs. Data Extraction.** MIA advantage is frequently used as a measure of information leakage (Shokri, 2022; Shokri et al., 2017; Mireshghallah et al., 2022a) and a proxy for measuring memorization (Carlini et al., 2021; Mireshghallah et al., 2022b), with recent attempts studying user-level leakage for the fine-tuning setting (Kandpal et al., 2023). However, the 'extractability' of training samples has recently become synonymous with memorization and is increasingly used to compare memorization across models (Biderman et al., 2023a; Carlini et al., 2023; Tirumala et al., 2022). Kandpal et al. (2022) investigated the impact of factors such as training data deduplication on extractability in a similar vein to our work on MIA. With extraction, a prefix is used as a prompt to measure the memorization of a sequence by comparing the resulting generation against the suffix. Both MIA and extraction are useful techniques for studying leakage in models, but rely on different assumptions and reveal different types of leakage risks. While MIAs require knowledge of candidates and only reveal directly which of those candidates are included in the training data, extraction requires knowledge of sufficient-length prefixes to perform extraction and additional measures to determine if extracted texts are valid.

## 3  Membership Inference on LLMs is Difficult

We perform a large-scale evaluation of five state-of-the-art MIAs (§2) on a range of LLMs with up to 12B parameters and diverse benchmarks. For the reference-based attack in Table 1 and all following experiments, we use STABLELM-BASE-ALPHA-3B-V2 as the reference model (determined empirically in §3.1). Code is available at `http://github.com/iamgroot42/mimir`.

**Target models.** We primarily target the PYTHIA model suite, including (1) five models of **PYTHIA** (Biderman et al., 2023b) with 160M, 1.4B, 2.8B, 6.7B, and 12B parameters [2], trained on the original Pile data (Gao et al., 2020), and (2) five models of **PYTHIA-DEDUP** (Biderman et al., 2023b) with the same parameter counts as PYTHIA but trained on the deduplicated Pile data. We also experiment with the GPT-NEO family to validate our findings with a different model family, observing similar trends in most domains (see Appendix A.6).

**Datasets.** We use seven diverse data sources included in the Pile: general web (Pile-CC), knowledge sources (Wikipedia), academic papers (PubMed Central, ArXiv), dialogues (HackerNews), and specialized-domains (DM Math, Github). We also perform experiments over the entire Pile. Members and non-members for each data source are sampled from

---

[2]We include results for the 70M variant in Table 1 for completeness over the model suite, but choose to focus mainly on the larger models for the later experiments.

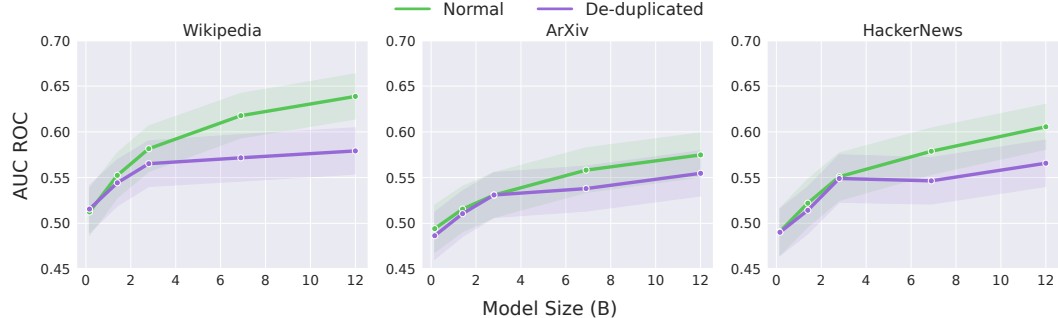

Figure 1: MIA performance as model size increases for the reference-based attack over select domains. We also plot the AUC ROC trajectory against the non-deduped Pythia suite for comparison. **Increasing model size slightly boosts MIA performance while deduplication decreases performance**. Other attacks follow similar trends (Appendix Figure 12).

the train and test sets of the Pile, respectively. Gao et al. (2020) decontaminated the Pile test set against the training set at a document level; nonetheless, to be more rigorous, we perform additional deduplication following Groeneveld et al. (2023). We additionally sample from documents greater than 100 words, and truncate samples up to 200 words. Refer to Appendix A.3 for further details.

**Evaluation metrics.** We primarily report **AUC ROC** for our evaluations, and additionally record **TPR@low%FPR** (Carlini et al., 2022) to assess performance in high-confidence settings. We visualize the 95% confidence interval for AUC ROC scores via shaded regions.

## 3.1 Main Results

**MIAs perform near random.** Table 1 shows that all existing MIAs perform near random for most domains[3]. No single MIA or target model demonstrates attack AUC above 0.6, with the exception of Github domain (see Appendix B.3 for discussion). Overall, the reference-based attack performs best, although there are a few settings where other attacks perform better, e.g., Min-$k$% Prob on Pile CC for the 160M PYTHIA-DEDUP model. Marginal differences in performance across MIAs make it hard to single out an overall *best* attack.

MIA performance tends to increase with the target model size (Table 1, Figure 1), in agreement with prior work (Shi et al., 2023; Li et al., 2023a; Watson et al., 2022). This is likely because larger models are more prone to overfitting the training data (Nakkiran et al., 2021). We also find deduplication of the training data reduces MIA performance (Figure 1), confirming the findings from Kandpal et al. (2022).

**Difficulty in Choosing a Reference Model.** We ablate the choice of reference models in the reference-based attack in Appendix A.5. In summary, (1) most reference models yield poor performance, with STABLELM-BASE-ALPHA-3B-V2 being the best overall, and (2) even aggregating all reference models performs poorly. In general, we find choosing the right reference model for a target LLM challenging and largely empirical. A reference model should be trained on the data that is same-distribution but largely disjoint from the training data of the target model. However, this assumption is hard to impose at the scale of pre-training corpora; common practice is to collect all the data available on the web, leading independently collected datasets to naturally overlap with each other.

---

[3]Similar trends for TPR@1%FPR. See Table 11 in Appendix.

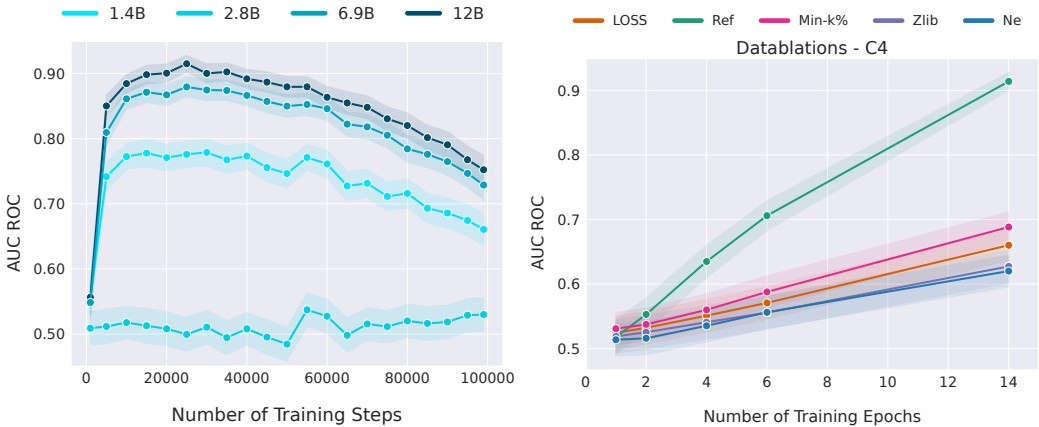

Figure 2: (Left) Reference-based attack performance as the amount of training data seen, measured in the number of training steps, increases across 1 epoch of the deduplicated Pile. In general, **performance spikes greatly before gradually decreasing as the amount of training data seen increases**. Other attacks (Figure 13, Appendix) follow similar trends. (Right) MIA performance on target model DATABLATIONS as the number of effective epochs increases via increasing epoch count. **Performance increases linearly with the number of effective epochs**. See Figure 10 for results on SILO.

## 3.2 Why is MI Challenging against LLMs

We identify several key factors that may contribute to the decreased performance of MIAs on LLMs. Some factors are due to unique characteristics of practices in LLM pre-training (§3.2.1) while others are due to inherent ambiguity in MIA (§3.2.2).

### 3.2.1 Characteristics of LLMs

**Training Data Size.** Current state-of-the-art pretrained LLMs are trained with billions and trillions of tokens (Touvron et al., 2023a;b; Team, 2023). We hypothesize the *large pretraining corpora characteristic to LMs decreases MIA performance*, as larger pretraining datasets lead to better generalization (Hoffmann et al., 2022; Muennighoff et al., 2023).

We employ the PYTHIA-DEDUP model suite's intermediate checkpoints to assess the impact of different amounts of training data. While keeping non-members fixed, we sample members for each checkpoint from its most recent 100 steps to remove the impact of the recency bias of the members.[4] See Appendix A.3.1 for details.

MIA performance generally starts as near-random, then rapidly increases within the next few thousand steps, before decreasing across successive checkpoints[5] (Figure 2, left). We speculate the initial low performance is due to the model warming up in training, with high losses across both member and non-member samples. We believe the following rapid rise and then gradual decline in performance are because the data-to-parameter-count ratio is smaller early in training and the model may tend to overfit (Yeom et al., 2018), but generalizes better as training progresses, in line with observations in existing work (Nakkiran et al., 2021).

**Number of Training Epochs.** It is standard practice to pre-train LLMs for around one epoch, given the scale of data and their tendency to overfit quickly (Muennighoff et al., 2023; Komatsuzaki, 2019). Previous MIA works that demonstrate attack effectiveness consider

---

[4]Using recently seen members elevates MIA performance noticeably, but doesn't disrupt the impact of increasing training data size. See Appendix C.1 for the impact of the recency of data seen in training.

[5]PYTHIA-DEDUP-2.8B stands apart with a performance trajectory that is consistently near-random. Previous work also observes unexplainable behavior for this model (Biderman et al., 2023a).

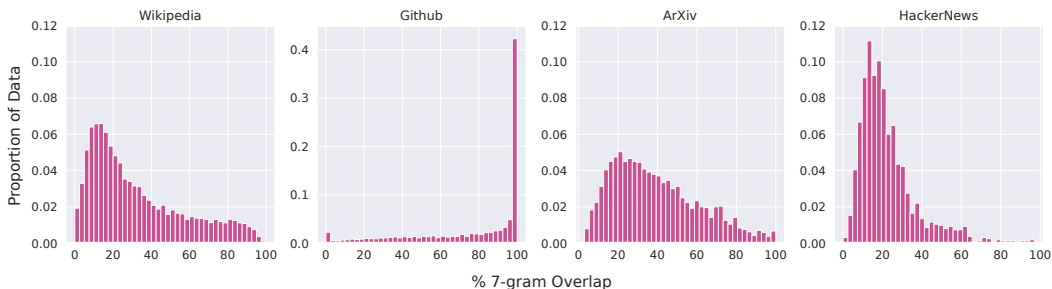

Figure 3: Natural distributions of 7-gram overlap of non-member data over select domains. Github has a considerably higher overlap than other domains.

supervised fine-tuning or masked LM pre-training (Nakamura et al., 2020; Lehman et al., 2021; Mireshghallah et al., 2022a;b), where models are trained for more than 10 epochs. We explore how the ***near-one epoch training of LLMs leads to decreased MIA performance***.

To verify this hypothesis, we perform MIA against the **Datablations** suite (Muennighoff et al., 2023), consisting of models trained on subsets of C4 (Raffel et al., 2019) train data for varying numbers of *epochs* (see Appendix A.2 for model details). We also experiment with **SILO** (Min et al., 2023); see Appendix C.2 for SILO results.

Increasing the number of effective epochs corresponds to an increase in attack performance (Figure 2, right). While Muennighoff et al. (2023) shows training for multiple epochs helps improve performance, our results suggest that such multi-epoch training (and/or large upsampling factors) can increase training data leakage.

### 3.2.2 Inherent Ambiguity in MIA

Natural language documents commonly have repeating text—even with the best efforts in decontamination and deduplication. These include common phrasings and quotes, natural use of similar texts, and syntactical similarities inherent to specific domains. This leads to substantial text overlap between members and non-members, which motivates the following hypothesis: ***higher overlap between members and non-members increases MIA difficulty***.

We quantify overlap using the percentage of $n$-gram overlap, defined as[6]: For a non-member sample $\mathbf{x}$ consisting of $m$ words such that $\mathbf{x} = x_1 x_2 ... x_m$ and an $n$-gram in $\mathbf{x}$ defined as a continuous substring $x_i...x_{i+n-1}$ , the $n$-gram overlap of $\mathbf{x}$ on training dataset $D$ is

$$\frac{1}{m-n+1} \sum_{i=1}^{m-n+1} \mathbb{1}\{\exists y \in D : x_i...x_{i+n-1} \in y\}$$

For a given non-member sample, this gives us the percentage of $n$-grams in the non-member that can be found in at least one member sample. We first compute the percentage of 7-gram overlap for non-members against the entire Pile training (member) set (Figure 3)[7]; see Appendix B.1 for implementation details. We observe high $n$-gram overlap with training data for a substantial portion of non-members; e.g., the Wikipedia, ArXiv, and PubMed Central domains have average 7-gram overlaps of 32.5%, 39.3%, and 41.0%, respectively. Domains such as GitHub, DM Mathematics, and FreeLaw see even higher overlap, with mean 7-gram overlap of 76.9%, 72.8%, and 62.3%, respectively.

High $n$-gram overlap suggests that substrings of non-members may be seen exactly during training, which makes the distinction between members and non-members even less clear. To verify our hypothesis, we resample non-members ensuring $\leq 20\%$ $n$-gram overlap with members, and report MIA performance (Table 2). While this step is designed to more strictly

---

[6]Non-members can still have 100% $n$-gram overlap without being members as $n$-grams can appear in different member samples. However, this is increasingly unlikely for larger $n$.

[7]Figure 14 shows $n$-gram overlap distributions for other $n$.

| Domain | Wikipedia | | Github | | PubMed Central | | Pile CC | | ArXiv | |
|---|---|---|---|---|---|---|---|---|---|---|
| | ORIG | 7-GRAM | ORIG | 7-GRAM | ORIG | 7-GRAM | ORIG | 7-GRAM | ORIG | 7-GRAM |
| LOSS | .516 | .666 | .678 | .878 | .506 | .780 | .516 | .574 | .527 | .787 |
| Ref | .579 | .677 | .559 | .615 | .559 | .595 | .582 | .644 | .555 | .715 |
| min-$k$ | .517 | .644 | .683 | .890 | .512 | .792 | .521 | .578 | .530 | .734 |
| zlib | .524 | .631 | .690 | .908 | .506 | .772 | .517 | .560 | .521 | .780 |
| Ne | .520 | .612 | .660 | .877 | .497 | .737 | .514 | .566 | .519 | .773 |

Table 2: Comparison of MIA performance over select domains with varying non-member sets at $\leq 20\%$ $n$-gram overlap threshold for $n = 7$, as well as the natural non-member set. Target model is PYTHIA-DEDUP-12B and AUC ROC reported. **Strict $n$-gram overlap thresholding results in higher performance**.

eliminate instances of non-member records that may overlap with training records, it also introduces an explicit drift between member and non-member distributions by selecting non-members that are most "unlike" training records. We clarify that this step is not a suggestion for researchers to alter their benchmarks; such a processing step drifts away from the standard membership inference game (Yeom et al., 2018).

**Results.** MIAs perform significantly better as the non-member distribution concentrates towards lower $n$-gram overlap and further diverges from the natural $n$-gram overlap distribution of non-members from the training distribution e.g., .516→.666 in Wikipedia, .690→.908 in Github, and .512→.792 in PubMed Central for various attacks. This is intuitive as the target model is likely to assign a lower likelihood to non-members further from its training data, making members and non-members more distinguishable. Note that decreasing the $n$-gram overlap threshold, especially for smaller $n$, pushes the setting closer to distribution inference (Suri & Evans, 2022), since the distributions of 'member' and 'non-member' records are no longer the same. We further discuss outlier behavior in Appendix B.

We note that $n$-gram overlap is an intrinsic property of natural language rather than a problem of the Pile train-test split. These splits are already deduplicated at a document level, following standard practice in decontamination (Gao et al., 2020; Brown et al., 2020). Nonetheless, repeating texts across distinct documents are fundamental and natural properties of domain data. We also note that $n$-gram overlap distribution analysis can help assess how representative of a target domain a set of candidate non-members is when constructing MIA benchmarks. Ultimately, we highlight the need to consider qualities of the data domain, e.g., $n$-gram overlap, and understand their potential impact on MIA performance.

## 4 Importance of Candidate Set Selection

In contrast to our findings in §3, recent works report state-of-the-art MIAs achieving > .7 AUC ROC on pretrained LLMs (Shi et al., 2023; Meeus et al., 2023). We investigate how the non-member candidate selection methods in these works can result in an inherent but likely unintended distribution shift between members and non-members as one such reason for the observed performance differences.

**Experimental Setup.** Prior work distinguishes members and non-members of a target domain based on the knowledge cutoff date of the target model, with members coming before and non-members coming after the cutoff. We construct similar experimental settings under two domains also used in earlier works: Wikipedia and ArXiv.

We follow the same setup as §3 on the Wikipedia domain, but replace the non-member set with samples from the entire RealTimeData WikiText dataset (Li et al., 2023c). The Pile members are sampled from articles in a Wikipedia dump from before March 2020 (Gao et al., 2020), whereas non-members consist of Wikipedia articles created from August 2023 and onwards. For the ArXiv domain or further details on Wikipedia, see Appendix A.3.2.

| # Params | Temporal Wiki | | | | |
|---|---|---|---|---|---|
| | LOSS | Ref | min-k | zlib | Ne |
| 160M | .643 | .602 | **.648** | .541 | .600 |
| 1.4B | .653 | **.705** | .682 | .572 | .603 |
| 2.8B | .667 | **.754** | .701 | .593 | .615 |
| 6.9B | .675 | **.788** | .714 | .601 | .620 |
| 12B | .680 | **.796** | .719 | .607 | .626 |

Table 3: AUC-ROC on the temporally shifted Wikipedia benchmark across various MIAs. Target models are the PYTHIA-DEDUP suite models. For each model, the highest score across MIAs is bolded.

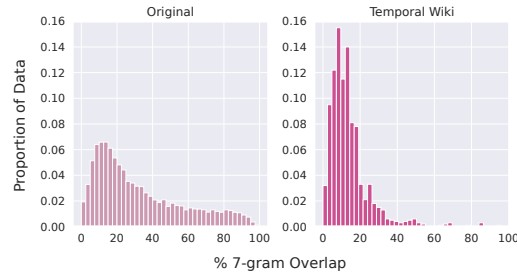

Figure 4: Distribution of 7-gram overlap for the original and temporally-shifted non-members.

**Results.** Table 3 demonstrates that the temporally shifted settings yield MIA performances significantly higher than when members and non-members are from the same temporal range. Figure 6 (Appendix) also demonstrates that MIA performance generally increases as non-members are further temporally shifted from member data. We speculate this follows from changes in language such as the introduction of new terminology and ideas over time.

*Temporal Shift as Change in n-gram Overlap.* We interpret temporal shift as a change in $n$-gram overlap distribution between the original and temporally shifted non-members. Figure 4 demonstrates that the distribution of 7-gram overlap of such shifted data concentrates at lower overlap percentages compared to their natural counterparts. The natural Wikipedia non-members have an average 7-gram overlap of 39.3%, whereas for the temporally shifted Wikipedia non-members it is 13.9%. See Appendix B.4 for temporal ArXiv discussion.

In general, when aiming to assess MIA performance, we advise estimating how representative a sample non-member set is of the member domain by comparing its $n$-gram overlap distribution with that of a left-out sample set from the pretraining corpora. Particularly, if the distribution of the candidate set is noticeably shifted towards lower $n$-gram overlap compared to the left-out member sample set, the candidate non-member set may not be representative of the member distribution from the target domain and potentially high MIA performances should be carefully examined. Closer inspection (Table 5, Appendix) reveals the extent of such over-estimation; decision thresholds derived using temporally-shifted non-members end up testing for temporal shift rather than membership. We note that distinguishing between members and temporally shifted non-members is a realistic inference game with practical implications but differs from the classical MI game as temporally-shifted data may belong to a different distribution. Furthermore, as we aren't able to reproduce the experimental settings of prior works (Shi et al., 2023; Meeus et al., 2023) for distributional shift analysis, it is inconclusive if their differing results are solely due to the temporal shift between member/non-member samples (see Appendix A.3.2 for details).

## 5 Revisiting Membership

The discussion of high $n$-gram overlap of non-members with members raises the question of whether exact membership is always useful concerning information leakage. The definition of membership in the standard MI game treats only records seen exactly during training as members, e.g., for language models, substrings appearing exactly in the training corpus. However, this may be at odds with what adversaries and privacy auditors care about when concerning information leakage. For generative models especially, guessing the membership of some sample via other sufficiently close samples can be useful. For example, any paraphrase of "Product launch in Q2, 2025" may be relevant as long as it preserves information regarding the product launch timeline, even if the paraphrase has a significant lexical difference, e.g., "Launching product in Q2, 2025" or "Q2 2025 release". Note that standard notions of Differential Privacy (Dwork et al., 2006) do not immediately protect against such cases, since records being tested for membership are not, in the literal sense,

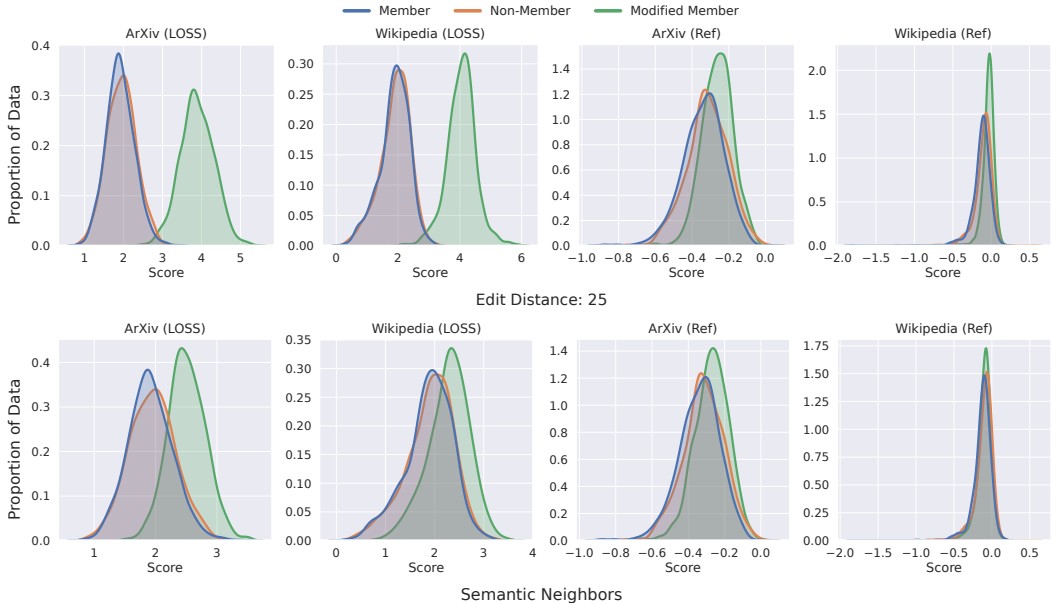

Figure 5: Distribution of scores for LOSS and Reference-based attacks for members, non-members, and modified members across ArXiv and Wikipedia domains. (Top) Modified members generated by random token replacement for edit distance 25. (Bottom) Modified members generated by replacing 5% of tokens with semantically similar tokens.

| Domain | 1% | | | 5% | | | 10% | | | Domain | LOSS | | | Ref | | |
|---|---|---|---|---|---|---|---|---|---|---|---|---|---|---|---|---|
| | 1 | 10 | 25 | 1 | 10 | 25 | 1 | 10 | 25 | | 1% | 5% | 10% | 1% | 5% | 10% |
| ArXiv | 0.1 | 0.0 | 0.0 | 0.3 | 0.1 | 0.1 | 0.7 | 0.3 | 0.2 | ArXiv | 0.0 | 0.8 | 2.5 | 0.7 | 1.9 | 4.0 |
| Wikipedia | 0.0 | 0.0 | 0.0 | 0.2 | 0.1 | 0.1 | 0.6 | 0.4 | 0.1 | Wikipedia | 0.0 | 0.5 | 2.3 | 0.4 | 3.0 | 8.2 |

Table 4: FPR (%) on modified members (treated as non-members) when using a score threshold that achieves a 1, 5, or 10% FPR on the original member and non-member data for ArXiv and Wikipedia domains. (Left) Results for lexically similar modified members at edit-distances $n = \{1, 10, 25\}$. Reference-based attack is shown. For LOSS attack, all FPR values are 0 across all tested FPR thresholds and values of $n$. (Right) Results for semantically close modified members. LOSS and Reference-based attack reported.

members. We explore two methods of constructing such "sufficiently close" samples as an initial step in studying approximate membership definitions.

**Lexical Distance.** We first experiment with creating modified member samples by replacing $n$ random tokens in a given sample with tokens randomly sampled from the model's vocabulary. We do so for $n = \{1, 10, 25\}$ (20 trials per $n$) and visualize the distribution for MIA scores using LOSS and Reference-based attacks (Figure 5, top). The LOSS attack yields distinct loss distributions between the modified members and original member/non-members, suggesting that the model is sensitive to out-of-place random tokens even for lightly perturbed member samples. The Reference-based attack, on the other hand, has a distribution of modified members much closer to both members and non-members, likely due to the reference model calibration accounting for the complexity introduced by the random tokens. This further reinforces the ambiguity of such samples—should they be considered members or non-members?

We also compute the thresholds corresponding to certain FPRs for actual member and non-member data and use these thresholds to compute the FPR on the modified members. We consider these modified members as "non-members", which they are with regards to

exact match[8]. Table 4 (left) shows that these modified members have extremely low FPRs for edit distance as low as $n = 1$, suggesting that these samples would be classified as non-members by the MIA, even though from the perspective of information leakage such a sample is effectively a member. We highlight the need to rethink membership for samples with extremely low lexical distance from actual training members, though even membership at higher lexical distances is important with respect to what information is still leaked in the unperturbed portions.

**Semantic Distance.** While a small edit distance suggests closeness in meaning, a higher edit distance does not necessarily imply loss of semantics. We compute MIA scores for neighbors generated for member samples as part of the Neighborhood attack (see Appendix A.4) for the Wikipedia and ArXiv benchmarks and repeat the above pipeline. Visualizing the scores shows how the modified members are not too far from original member score distributions, especially for the Reference attack (Figure 5, bottom). We repeat the same FPR experiment as edit-distance-based modified members. While the FPR for these semantically close records is noticeably higher than records close by edit distance, the false positive rates are still low (Table 4, right). Again, these results suggest that semantically close members would be classified as non-members even though they may be as useful as actual members depending on the inference goal and the semantic information preserved.

While it is not surprising that semantically close neighbors have MIA scores more similar to actual member samples than randomly-replaced tokens, it is clear that an ideal distance function should combine the benefits of lexical distance and semantics in defining a membership neighborhood. Such observations also motivate a fully semantic MI game, where a neighbor member may be defined by its proximity to an exact member in a semantic embedding space. This may provide a clearer interpretation of knowledge leakage than lexical matching, especially when samples naturally have high lexical (i.e., $n$-gram) overlap.

## 6 Conclusion

We shed light on the difficulty of membership inference against LLMs from the lens of an adversary. Our results suggest two possibilities: (1) data does not leave much of an imprint, owing to characteristics of the pre-training process at scale, such as large datasets and single-epoch training, and (2) the similarity between in and out members (which we demonstrate via $n$-gram overlap), coupled with huge datasets, makes this distinction fuzzy. While MIA performance could improve via stronger attacks (Casper et al., 2024), the second factor requires rethinking the membership game itself. Our empirical results suggest that both of these might be confounding factors while measuring leakage. The membership inference game needs to be extended for such generative models to better align with information leakage that adversaries and auditors may care about, such as user-level leakage (Kandpal et al., 2023) and PII (Lukas et al., 2023). In the meanwhile, special care should thus be taken to avoid unintentional distributional shifts while constructing non-members for MIA benchmark construction.

## Acknowledgements

We are grateful to Nicholas Carlini for comments on early drafts of this paper. This research is supported in part by DARPA SemaFor Program No. HR001120C0123, and grants from the National Science Foundation through the AI Institute for Agent-based Cyber Threat Intelligence and Operation (#2229876), Center for Trustworthy Machine Learning (#1804603) and the DARPA MCS program through NIWC Pacific (N66001-19-2-4031).

This work is also supported in part by the Office of the Director of National Intelligence (ODNI), Intelligence Advanced Research Projects Activity (IARPA), via the HIATUS Program contract #2022-22072200004. The views and conclusions contained herein are those of

---

[8]We perform token replacements at random with random tokens, so it is unlikely that these edited members are also actually members.

the authors and should not be interpreted as necessarily representing the official policies, either expressed or implied, of ODNI, IARPA, or the U.S. Government. The U.S. Government is authorized to reproduce and distribute reprints for governmental purposes notwithstanding any copyright annotation therein.

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

# A  Implementation details

## A.1  MIMIR

We release our codebase as a Python package (available at `http://github.com/iamgroot42/mimir`), complete with documentation and tests, for implementing and evaluating membership inference attacks for language models. The data used in our experiments is also available via HuggingFace (`https://huggingface.co/datasets/iamgroot42/mimir`). Some features of the package include:

- A base attack class that provides bare-bones code and helper functions that can be easily used in implementations of both existing and new attacks.
- Data processing utilities to filter and cache data for membership evaluation, from both provided and available sources.
- Support for a vast array of models that can be used as target or reference models.

The entire codebase works with modular configuration files, allowing multiple experiments to be run simultaneously with no edits to the code itself. We used Python 3.9.7 for our experiments, with PyTorch 2.0.1. Our experiments were executed on a mix of machines, with GPUs ranging from RTX6k to A100.

## A.2  Additional Target Model Details

**PYTHIA-DEDUP.** Both the PYTHIA and PYTHIA-DEDUP model suites provide intermediate checkpoints for each model. For experiments targeting the PYTHIA-DEDUP model, as the PYTHIA-DEDUP model is trained for greater than 1 epoch, we select the checkpoint that most closely matches the one epoch mark over the deduplicated Pile. We decide this is checkpoint 'step99000'. For experiments targeting the non-deduped PYTHIA models, we use the final checkpoint, which sees just under one ($\approx 0.9$) epoch of the original Pile.

**SILO.** The models from the SILO suite (Min et al., 2023) consist of 1.3B-parameter transformer LMs based on the OpenLM implementation of the LLaMA architecture (Touvron et al., 2023a). These are trained for multiple epochs on the Open License Corpus, which consists of permissively-licensed text data classified as either **public domain** (PD) texts, **permissively licensed software** (SW), or under an **attribution license** (BY). We target the SILO-PDSW model (alongside its intermediate checkpoints) trained on only texts classified as PD or SW for domains contributing less than 5% of the data upsampled by a factor of 3x (which includes HackerNews and DM Mathematics).

**DATABLATIONS.** The DATABLATIONS suite (Muennighoff et al., 2023) is a large collection of models trained to study scaling laws in data-constrained regimes. They vary in the extent of data repetition and compute budget, ranging up to 900 billion training tokens and 9 billion parameters. For the epoch experiment, we choose the 2.8B-parameter subset of models, with each seeing a total of 55B tokens from the C4 dataset across their training runs. These models vary in the number of epochs their training subset is seen, ranging from one to 14 epochs. They also offer a model trained for 44 epochs, which we decided to leave out of evaluation.

**GPT-NEO.** is a collection of 125M-, 1.3B-, and 2.7B-parameter models of similar architecture to the GPT-3 model family. These models are trained on the Pile for about 300B tokens, similar to the PYTHIA suite. This model suite is a precursor to the GPT-NEOX (Andonian et al., 2023) model architecture, which PYTHIA-DEDUP and PYTHIA are built on. Noticeable differences include the tokenizer used per model suite, with the GPT-NEOX allocating additional tokens to whitespace characters, as well as intended training settings, with GPT-NEO geared towards TPU training and GPT-NEOX GPU training.

**OLMO.** The OLMO model suite (Groeneveld et al., 2024) is a suite of open language models trained on the DOLMA(Soldaini et al., 2023) dataset. OLMO models currently available

include 1B- and 7B-parameter variants trained on 3T and 2.5T tokens, respectively. While our preliminary results just target the final checkpoints, the OLMO suite is similar to the PYTHIA suite in that intermediate checkpoints and exact training order are fully open.

### A.3 Benchmark Details

We sample 1,000 members and non-members from each target domain from the Pile train and test sets, respectively. We do the same for the aggregate Pile experiment, except we sample 10,000 members and non-members each from the complete Pile train and test sets. We sample documents greater than 100 words and truncate them up to 200 words from the beginning to create our benchmark examples. Previous work (Shi et al., 2023) observes that sample length correlates with performance, so we bound the sample length to reduce its impact while picking a reasonable threshold so that our samples are likely to contain ample signal. While further increasing the length of samples could yield greater MIA performance, such an experiment is orthogonal to the ones we conduct; inherent differences in LLM training and MI evaluation would still impact evaluation on longer texts.

We follow the same pipeline when generating the benchmark for targeting the DATABLA-TIONS models, picking members and non-members from the C4 train and validation sets, respectively.

For our additional decontamination on Pile benchmarks, we follow Groeneveld et al. (2023), which uses a bloom filter to check for $n$-gram inclusion. We keep the default filtering settings of $n = 13$ and a threshold of $\leq 80\%$ overlap. Further details about setting up the bloom filter can be found in Appendix B.1

#### A.3.1 Training Data Size Benchmarks

For each model, we pick checkpoints every 5000 steps ending at step 95000, with each step corresponding to 1024 samples of length 2048 tokens. We also include checkpoints at step 1000 and step 99000, the closest checkpoint to the step where one full epoch of the deduplicated Pile was seen. For each checkpoint, we use the same non-member set for evaluation consisting of 1000 samples sampled from the entire Pile test set. We then construct a member set for each checkpoint[9]: for the checkpoint at step $n$, we sample 1000 random samples from documents seen within the range step $\{n - 100, n\}$. EleutherAI provides random seeding for deterministic training data order across the PYTHIA-DEDUP training runs, which we use to determine the seen document order. This allows us to determine which documents to sample from for a given step range. For both members and non-members, we sample with the same criterion as the general experiments above.

#### A.3.2 Temporal Benchmarks

For the temporal Wikipedia benchmark non-members, we collect samples from the Real-TimeData "**wikitext_latest**" dataset (Li et al., 2023c). This yielded Wikipedia articles created between the week of August 12, 2023 till the week of January 8, 2024 [10]. We then follow Pile processing steps by simply appending the article titles to the front of each respective article with a "\n\n". Members are sampled from the Wikipedia subdomain of the Pile training set. Members and non-members are then sampled with the same criterion as in the general experiments.

For the temporal ArXiv benchmarks, the member set for each benchmark is fixed and sampled from the ArXiv subdomain of the Pile training set, which consists of papers posted prior to July 2020 (Gao et al., 2020). For non-members, we use the ArXiv API again following

---

[9]Ideally, the member set should be fixed, which could be done by performing multiple training runs and injecting the fixed member set at various steps. However, this is computationally expensive. Furthermore, because of how the data is shuffled, we'd expect the difficulty of the member set to be reasonably consistent across our samples

[10]Note that while the articles are created in the recent time frame, the contents of the Wikipedia page aren't necessarily about recent topics, people, or events

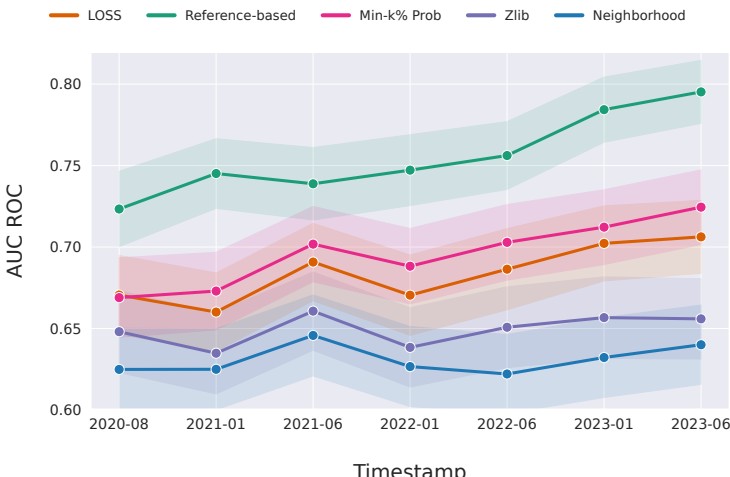

Figure 6: MIA performance across benchmarks where non-member data is selected from ArXiv preprints created during increasingly later months past the target model's knowledge cutoff. Timestamps are formatted as year-month. The target model is PYTHIA-DEDUP-12B. In general, **MIA performance increases as the temporal shift of non-members increases**

Li et al. (2023c) to collect ArXiv preprints from specific months: August 2020, January 2021, June 2021, January 2022, June 2022, January 2023, and June 2023 [11]. We then apply the same processing steps used in the Pile (Gao et al., 2020). This mainly involves converting the latex sources for a given preprint into a single Markdown file, and then filtering out documents such as those with conversion errors. For each month range, we sample non-members from processed files in the given date range. By sampling non-members from successively later time ranges after the Pile ArXiv cutoff date, we also seek to explore how greater temporal shift impacts MIA performance. We again sample both members and non-members with the same criterion as in the general experiments.

**Difficulties in reproducing and analyzing existing works.** While we follow a similar method of non-member candidate selection for our temporal experiments as prior works such as Shi et al. (2023) and Meeus et al. (2023), we were unable to reproduce their settings for analysis for two main reasons: 1) their exact non-member candidates and/or pipelines to reproduce the non-member candidate selection are unavailable or 2) target models used such as LLAMA (Touvron et al., 2023a) do not release their training data to conduct $n$-gram or other distributional analyses.

### A.4  Attack Details

MIAs consider a target model $\mathcal{M}$, which outputs a probability distribution of the next token given a prefix, denoted as $P(x_t|x_1...x_{t-1}; \mathcal{M})$. Their goal is to model $f(\mathbf{x}; \mathcal{M})$, which outputs a score for target sample $\mathbf{x} = x_1...x_n$ with $n$ tokens. This score is then thresholded to determine the target sample's membership in the training data of $\mathcal{M}$.

**LOSS** (Yeom et al., 2018; Carlini et al., 2019) considers the model's computed loss over the target sample:

$$f(\mathbf{x}; \mathcal{M}) = \mathcal{L}(\mathbf{x}; \mathcal{M}).$$

**Reference-based** (Sablayrolles et al., 2019; Watson et al., 2022) attacks assume access to a reference model $\mathcal{M}_{\text{ref}}$, another LM trained on a disjoint set of training data drawn from a similar distribution. In practice, an assumption of disjoint training data is impractical.

---

[11]This slightly differs from the Pile ArXiv data collection, which uses the ArXiv bulk access through S3. However, we believe both ArXiv bulk access and API should yield the preprints in the same manner regardless.

| Thresholding Benchmark | 1% | | | | 5% | | | | 10% | | | |
|---|---|---|---|---|---|---|---|---|---|---|---|---|
| | LOSS | Ref | min-$k$ | zlib | LOSS | Ref | min-$k$ | zlib | LOSS | Ref | min-$k$ | zlib |
| 2020-08 | 3.2 | 4.2 | 4.5 | 3.7 | 12.6 | 13.4 | 13.5 | 13.8 | 24.1 | 23.3 | 24.6 | 20.2 |
| 2021-01 | 3.7 | 3.9 | 3.5 | 3.5 | 11.4 | 15.8 | 13.5 | 10.4 | 21.7 | 27.0 | 24.6 | 17.5 |
| 2021-06 | 3.2 | 4.2 | 5.7 | 5.4 | 14.4 | 16.0 | 15.7 | 13.6 | 25.5 | 25.5 | 29.5 | 23.0 |
| 2022-01 | 4.5 | 4.2 | 5.3 | 4.1 | 14.4 | 16.3 | 14.6 | 12.7 | 24.5 | 27.0 | 28.7 | 22.0 |
| 2022-06 | 2.8 | 3.9 | 3.1 | 2.5 | 10.3 | 18.1 | 13.1 | 10.7 | 23.4 | 27.8 | 25.4 | 20.6 |
| 2023-01 | 2.9 | 8.5 | 3.5 | 3.1 | 11.9 | 23.5 | 13.5 | 10.9 | 25.0 | 36.1 | 26.3 | 21.9 |
| 2023-06 | 5.8 | 9.4 | 5.5 | 5.8 | 15.6 | 22.7 | 19.1 | 14.1 | 26.3 | 37.3 | 27.8 | 22.2 |
| Temporal Wiki | 9.8 | 7.5 | 10.3 | 7.9 | 23.8 | 22.8 | 24.3 | 17.6 | 30.0 | 34.1 | 35.0 | 22.8 |

Table 5: FPR (%) on non-members from the Pile (original; not temporally shifted) on various attacks when using a score threshold that achieves a 1, 5, or 10% FPR on the temporally-shifted ArXiv (for varying levels of temporal shift) and Wikipedia benchmarks. The target model is PYTHIA-DEDUP-12B. **FPRs on the original non-members are much higher then the thresholded FPR on the temporally shifted benchmarks**, indicating that such thresholds may be moreso classifying temporal shift rather than member and non-members.

Empirically, using an LM that is different from $\mathcal{M}$ has been a reasonable choice and was used in prior work (Kandpal et al., 2022; Watson et al., 2022). The attack considers the membership score of the target sample by $\mathcal{M}$ relative to the membership by $\mathcal{M}_{ref}$ to calibrate the target model's score given a difficulty estimate through the reference model's score, with goals to improve precision and reduce the false negative rate. For our experiments, we use LOSS as the uncalibrated membership score such that, for the reference-based attacks,

$$f(\mathbf{x}; \mathcal{M}) = \mathcal{L}(\mathbf{x}; \mathcal{M}) - \mathcal{L}(\mathbf{x}; \mathcal{M}_{ref}).$$

This method exactly follows the method from Watson et al. (2022) and is also largely similar to the offline Likelihood Ratio attack (LiRA; Carlini et al. (2022)), although LiRA uses many reference models (often trained shadow models).

**Zlib Entropy** (Carlini et al., 2021) functions similarly to reference-based MIA, using the zlib compression size of a sample $\mathbf{x}$ as a local difficulty threshold per sample:

$$f(\mathbf{x}; \mathcal{M}) = \frac{\mathcal{L}(\mathbf{x}; \mathcal{M})}{\text{zlib}(\mathbf{x})},$$

where zlib($\mathbf{x}$) is the length in bytes of the zlib compressed sample.

**Neighborhood Attack** (Mattern et al., 2023) assumes access to a masking model, and operates by generating "neighbor" texts $\tilde{\mathbf{x}}$ to a given text sequence $\mathbf{x}$ by using the masking model to replace a percentage of randomly selected token spans while still maximizing the neighbor's likelihood. If the sample's loss is considerably lower than the neighbor's losses, the difference is attributed to the target model overfitting the sample, and the sample is considered a training member. Formally, we have

$$f(\mathbf{x}; \mathcal{M}) = \mathcal{L}(\mathbf{x}; \mathcal{M}) - \frac{1}{n} \sum_{i=1}^{n} \mathcal{L}(\tilde{\mathbf{x}}_i; \mathcal{M}).$$

We use BERT (Devlin et al., 2019) as our masking model of choice, with a masking percentage of 5%.

**Min-$k$% Prob** (Shi et al., 2023) is based on the intuition that non-member examples tend to have more tokens assigned lower likelihoods than member examples do. Given sample $\mathbf{x} = x_1, ..., x_n$ and hyperparameter $k$, let min-$k(\mathbf{x})$ be the set formed by the $k$% of tokens in $\mathbf{x}$ with minimum likelihood. We then have

$$f(\mathbf{x}; \mathcal{M}) = \frac{1}{|\text{min-}k(\mathbf{x})|} \sum_{x_i \in \text{min-}k(\mathbf{x})} - \log(p(x_i \mid x_1, ..., x_{i-1})).$$

We experiment with multiple different $k \in \{10, 20, 30, 40, 50\}$ as suggested in Shi et al. (2023), but settle on $k = 20$ for our experiments.

We compute the performance of each attack based on 1,000 bootstrap samples of the benchmark and report the average AUC ROC and TPR@low%FPR over the bootstraps.

**MIAs involving meta-classifiers.** While many recent works study MIAs that perform membership classification through meta-classifiers trained on features extracted from a ground-truth subset of member/non-member data, we primarily focus on blackbox attacks as access to such a subset can be difficult to guarantee in practice, especially as the inclusion of samples in pre-training corpora becomes more ambiguous as these corpora continue to expand.

### A.5 Reference Model Choices

We choose a diverse set of reference models to experiment with. For the aggregate method over all reference models, we take the average of the scores per reference model for a target sample[12]. We report results for our complete ablation on reference model choice in Table 6.

GPT-2 (Radford et al., 2019) is suite of pre-trained transformer trained on a large dataset of around 40GB of web text, likely overlapping with the Pile. We use the GPT-2-small variant with 124M parameters.

DISTILGPT2 (Sanh et al., 2019) is a smaller 82M-parameter model trained with the supervision of GPT-2-small using knowledge distillation.

OPT (Zhang et al., 2022) is a suite of open-sourced pre-trained transformers that are trained on a curated pre-training corpus including several datasets from the Pile, such as Wikipedia, DM Mathematics, and HackerNews. We use the 1.3B-parameter variant.

As mentioned in Appendix A.2, GPT-NEO (Black et al., 2021) is another suite of pre-trained transformers designed using EleutherAI's replication of the GPT-3 architecture. These models are trained on the full Pile for a similar amount of tokens as PYTHIA ($\sim$ 300B), though the data seen may not necessarily be in the same order as the PYTHIA models. We use the 1.3B-parameter variant.

SILO-PDSWBY (Min et al., 2023) is a 1.4B-parameter transformer pre-trained on all types of permissively licensed data in the Open License Corpus. The training data consists of certain Pile domains such as HackerNews and DM Mathematics.

LLAMA (Touvron et al., 2023a) is a collection of large, open-sourced pre-trained LMs ranging in size from 7B to 65B parameters. The pre-training corpus is on the scale of trillions of tokens, much larger than the Pile, and likely has significant overlap with the Pile. We use the 7B-parameter variant.

STABLELM-ALPHA-V2 (Tow, 2023) is a set of open-source pre-trained LMs also trained on a large pre-training corpus with trillions of tokens. Training is conducted in two stages, with the first stage seeing 1 trillion tokens of a mixture of data from sources such as RedPajama (Together AI, 2023) and the Pile, with an emphasis on refined web text. The second stage is trained on 100 billion tokens with a higher context length, increasingly sampling naturally long texts and adding the StarCoder (Li et al., 2023b) dataset. We use the 3B-parameter variant.

We also experiment with the non-deduped PYTHIA-DEDUP-1.4B model as a reference model to see how using a smaller version of the target model (same architecture and training data order) impacts reference-based attack performance (Carlini et al., 2021).

### A.5.1 Stablelm-Base-Alpha-3B-v2 Performance

We speculate that the slightly higher performance with STABLELM-BASE-ALPHA-3B-V2 as the reference model, even though its pre-training corpus has high overlap with the Pile, is

---

[12]Note that the scores over different reference models may not be directly comparable due to the reference models having different tokenizers. This may contribute to the poor performance of this naive ensembling method.

| # Params | Wikipedia | | | | | | | | Pile CC | | | | | | | |
|---|---|---|---|---|---|---|---|---|---|---|---|---|---|---|---|---|
| | GPT2 | DISTIL | OPT | NEO | SILO | LLAMA | STABLE | PYTHIA | GPT2 | DISTIL | OPT | NEO | SILO | LLAMA | STABLE | PYTHIA |
| 160M | .498 | .502 | .494 | .490 | .492 | .511 | **.515** | .480 | **.520** | .504 | .488 | .473 | .504 | .487 | .497 | .480 |
| 1.4B | .503 | .505 | .507 | .500 | .502 | .521 | **.544** | .476 | .523 | .507 | .513 | .500 | .516 | .504 | **.525** | .496 |
| 2.8B | .511 | .510 | .519 | .532 | .531 | .539 | **.565** | .526 | .526 | .509 | .521 | .499 | .520 | .510 | **.537** | .504 |
| 6.9B | .510 | .507 | .517 | .518 | .516 | .536 | **.571** | .501 | .538 | .520 | .542 | .525 | .531 | .530 | **.564** | .540 |
| 12B | .514 | .510 | .522 | .528 | .529 | .546 | **.579** | .517 | .548 | .525 | .555 | .538 | .541 | .545 | **.582** | .555 |

| # Params | PubMed Central | | | | | | | | ArXiv | | | | | | | |
|---|---|---|---|---|---|---|---|---|---|---|---|---|---|---|---|---|
| | GPT2 | DISTIL | OPT | NEO | SILO | LLAMA | STABLE | PYTHIA | GPT2 | DISTIL | OPT | NEO | SILO | LLAMA | STABLE | PYTHIA |
| 160M | .495 | .491 | .515 | .511 | .513 | .515 | **.516** | .497 | **.523** | .518 | .516 | .480 | .496 | .492 | .486 | .472 |
| 1.4B | .493 | .491 | .514 | .517 | .514 | .515 | **.530** | .503 | **.529** | .524 | .523 | .501 | .512 | .506 | .510 | .484 |
| 2.8B | .494 | .492 | .513 | .518 | .515 | .518 | **.536** | .500 | **.534** | .528 | .528 | .524 | .522 | .516 | .531 | .528 |
| 6.9B | .499 | .496 | .519 | .527 | .520 | .530 | **.552** | .526 | .540 | .532 | .534 | .539 | .531 | .528 | .538 | **.554** |
| 12B | .504 | .498 | .523 | .531 | .524 | .538 | **.559** | .533 | .546 | .538 | .541 | .555 | .540 | .538 | .555 | **.581** |

| # Params | DM Math | | | | | | | | HackerNews | | | | | | | |
|---|---|---|---|---|---|---|---|---|---|---|---|---|---|---|---|---|
| | GPT2 | DISTIL | OPT | NEO | SILO | LLAMA | STABLE | PYTHIA | GPT2 | DISTIL | OPT | NEO | SILO | LLAMA | STABLE | PYTHIA |
| 160M | .489 | .488 | .520 | .509 | .487 | .502 | **.523** | .514 | **.496** | **.496** | **.496** | .480 | .398 | .486 | .490 | .466 |
| 1.4B | .487 | .485 | .509 | .496 | .485 | .503 | **.512** | .496 | .508 | .509 | .511 | .496 | .401 | .504 | **.514** | .483 |
| 2.8B | .485 | .486 | **.511** | .503 | .483 | .500 | .504 | .509 | .521 | .522 | .529 | .534 | .421 | .521 | **.549** | .527 |
| 6.9B | .485 | .485 | **.510** | .499 | .484 | .502 | .508 | .497 | .525 | .526 | .534 | .536 | .436 | .531 | **.546** | .542 |
| 12B | .487 | .486 | **.514** | .504 | .485 | .502 | .512 | .503 | .534 | .533 | .545 | .559 | .453 | .545 | **.565** | .561 |

Table 6: The effect of the choice of a reference model to PYTHIA-DEDUP models across various domains. The reference model yielding the highest performance, per target domain and target model, is bolded. ROC-AUC values are reported.

because 1) larger target models[13] such as the PYTHIA-DEDUP-12B model may considerably overfit certain member samples and 2) the STABLELM-BASE-ALPHA-3B-V2 is trained on a much larger corpus, which helps it generalize well and achieve similar losses as the target model on the non-member data. As a result, member samples are more likely to have a greater magnitude of difference between the target and reference model losses compared to the difference between losses on non-members.

## A.6 Results with GPT-Neo models

We repeat our experiments with the GPT-NEO family of models. Table 7 demonstrates similar trends targeting the GPT-NEO models as seen when targeting the PYTHIA model family (Table 1, Table 12), such as MIA performance generally increasing as the target model size increases. In general, performance against the GPT-NEO models is similar, if not lower than, performance against the PYTHIA-DEDUP and PYTHIA models when comparing similarly sized variants. In some domains such as HackerNews, the best performing MIA differs between target models (Min-$k$% Prob for GPT-NEO, reference-based for PYTHIA-DEDUP), though marginally.

## A.7 Results with OLMo models

We perform preliminary experiments targeting the OLMO family of models. Benchmark construction is similar to what is detailed in Appendix A.3. However, we sample from domains that make up DOLMA (Soldaini et al., 2023), namely Wikipedia, C4, Reddit, Common Crawl, and peS2o (Soldaini & Lo, 2023). These are similar to domains used in Pile. We note that peS2o consists of both abstracts (s2ag) and full papers (s2orc), and evaluate them as separate domains. To get non-member, we use held-out DOLMA data from PALOMA (Magnusson et al., 2023).

Table 8 also demonstrates generally near-random performance trends, with both the 1B- and 7B-parameter model variants exhibiting similar performances. We speculate that, due to the incredibly large amounts of training data (3T tokens for 1B-parameter model, 2.5T tokens

---

[13]Also target models that are domain specific like DATABLATIONS or are trained on a less diverse corpus like SILO

| # Params | Wikipedia | | | | Github | | | | Pile CC | | | | Pubmed Central | | | |
|---|---|---|---|---|---|---|---|---|---|---|---|---|---|---|---|---|
| | LOSS | Ref | min-$k$ | zlib | LOSS | Ref | min-$k$ | zlib | LOSS | Ref | min-$k$ | zlib | LOSS | Ref | min-$k$ | zlib |
| 125M | .504 | **.511** | .492 | **.511** | .641 | .582 | .642 | **.660** | .495 | .492 | **.500** | .497 | .499 | **.506** | .502 | .499 |
| 1.3B | .510 | **.531** | .506 | .517 | .681 | .570 | .681 | **.696** | .500 | **.517** | .503 | .501 | .496 | **.499** | **.499** | .497 |
| 2.7B | .513 | **.545** | .513 | .519 | .699 | .570 | .700 | **.712** | .504 | **.531** | .507 | .506 | .498 | **.507** | .501 | .499 |

| # Params | ArXiv | | | | DM Math | | | | HackerNews | | | | The Pile | | | |
|---|---|---|---|---|---|---|---|---|---|---|---|---|---|---|---|---|
| | LOSS | Ref | min-$k$ | zlib | LOSS | Ref | min-$k$ | zlib | LOSS | Ref | min-$k$ | zlib | LOSS | Ref | min-$k$ | zlib |
| 125M | **.507** | .494 | .503 | .501 | .492 | **.522** | .493 | .484 | .489 | .480 | **.505** | .496 | .502 | **.507** | .505 | .505 |
| 1.3B | .511 | .506 | **.512** | .507 | .486 | **.511** | .491 | .481 | .499 | .500 | **.514** | .501 | .505 | **.514** | .509 | .507 |
| 2.7B | .515 | **.520** | .517 | .510 | .486 | **.509** | .492 | .481 | .502 | .512 | **.516** | .503 | .507 | **.519** | .511 | .509 |

Table 7: AUC ROC of MIAs against GPT-NEO across different datasets from the Pile. The highest performance across the different MIAs is bolded per domain. Similar to PYTHIA-DEDUP, **MIA methods perform near random ($<$ .55) in most domains**.

for the 7B-parameter model), performance across different model sizes begins to converge to near-random performance even with such distinct model sizes. Interestingly, the Reference-based attack using STABLELM-BASE-ALPHA-3B-V2 performs much worse than when used to calibrate the PYTHIA models, reinforcing the difficulty in finding suitable reference models for different LLMs. We also observe many settings where MIA performance is considerably less than .5, suggesting that the MIAs are more likely to predict members as non-members, and vice versa. More investigation is needed to understand such behaviors on specific domains such as s2ag from peS2o in the DOLMA data.

| # Params | Wikipedia | | | | C4 | | | | Reddit | | | |
|---|---|---|---|---|---|---|---|---|---|---|---|---|
| | LOSS | Ref | min-$k$ | zlib | LOSS | Ref | min-$k$ | zlib | LOSS | Ref | min-$k$ | zlib |
| 1B | .484 | **.510** | .495 | **.510** | .515 | .479 | **.520** | .513 | .464 | **.495** | .478 | .470 |
| 7B | .481 | .488 | .493 | **.500** | .516 | .499 | **.520** | .514 | .463 | **.501** | .480 | .469 |

| # Params | Common Crawl | | | | s2ag | | | | s2orc | | | |
|---|---|---|---|---|---|---|---|---|---|---|---|---|
| | LOSS | Ref | min-$k$ | zlib | LOSS | Ref | min-$k$ | zlib | LOSS | Ref | min-$k$ | zlib |
| 1B | .509 | .412 | **.517** | .511 | .449 | .376 | **.461** | .392 | .484 | .480 | **.500** | .463 |
| 7B | .498 | .410 | **.505** | .500 | .465 | **.483** | .475 | .406 | .491 | **.507** | .503 | .470 |

Table 8: AUC ROC of MIAs against OLMO across different datasets from the Dolma dataset. The highest performance across the different MIAs is bolded per domain.

# B $n$-gram Overlap Details and Takeaways

## B.1 Measuring $n$-gram Overlap

We create a bloom filter following Groeneveld et al. (2023). Due to the scale of the Pile training data and limited memory, we shard the bloom filter. In our construction, we split the training data in half, resulting in two bloom filter shards. Since each shard only sees half of the training data, to check for $n$-gram inclusion across the entire Pile, we check for containment in both of the sharded bloom filters, counting an $n$-gram included only if it is included in at least one of the bloom filters.

For each shard, we configure the bloom filter according to the data size such that the false positive rate of the bloom filter is less than 1% (0.6%). Then, for each document, we tokenize at the word level. We then add $n$-gramš to the filter by using a striding window over $n$ words at a time with a stride of 1. We use the same method of gathering $n$-gramš when checking the non-members for $n$-gram overlap.

## B.2 Reference-based Attack Performance

Table 2 shows that, interestingly, reference-based MIAs have a noticeably smaller increase in performance compared to non-referenced-based MIAs for domains such as GitHub or

PubMed Central under $n$-gram overlap thresholding. We speculate that, since numerous low $n$-gram overlap non-members are outliers to the relevant domain, these non-members will also be outliers to the similar/overlapping data seen by the reference model. As a result, even though these non-members may yield higher losses from the target model, we see similar high losses for the reference model as well, which makes the difference between target and reference model loss for non-members and members relatively less distinguishable compared to signals from the other attacks.

At the same time, domains like Pile CC do not see this dampened performance, likely because the 20% threshold in the case of Pile CC is not sufficient to select outliers, as samples from this domain have naturally low $n$-gram overlap. Another case where the reference-based attack seems to avoid this observation is in the temporally shifted non-member setting for both Wikipedia and ArXiv despite the temporally shifted non-members being more out-of-distribution relative to the Pile Wikipedia and ArXiv distributions, respectively. We speculate this is due to the reference model of choice, STABLELM-BASE-ALPHA-3B-V2, which has not only been trained on a corpus with high overlap with the Pile, but also trained on datasets that capture more recent data such as RedPajama-Data-1T (Together AI, 2023) which contains Wikipedia and ArXiv samples from a much more recent cutoff date (i.e., RedPajama uses the 2023-03-20 Wikipedia dump), allowing it to generalize better over the temporally shifted non-members and avoiding a shift towards higher losses that weaker or older reference models might experience.

Attacks like LOSS or Min-$k$% Prob do not utilize any external signal or difficulty calibration, and thus rely exclusively on signals from the target model for member classification. Calibration-based methods like zlib and reference-based attacks, on the other hand, account for the inherent "difficulty" of a seen sample. Thus, in situations where the non-member data is significantly out of domain, even for a reference model or calibration method, it is likely that the signals from the target model and difficulty calibration would cancel out, leading to a weakened MIA signal. On the other hand, difficulty calibration can further boost MIA signal in settings where the member data is inherently more likely to be memorized, such as in §3.2.1 where reference-based attacks yielded considerably higher MIA performance in low training data size and high effective epoch count settings, with performance being further amplified in the extremes of both settings. Thus, MIA baselines for new MIAs should include both kinds of methods: calibration-based and calibration-free. Having baseline coverage for both styles of MIA can help uncover inherent characteristics of the evaluation setting such as unintentional member/non-member distributional shift or overfitted target models that influence MIA performance and also paints a holistic picture with regards to what MIAs are most suitable for specific attack settings.

### B.3   GitHub as an Outlier

As seen in Table 1, MIA performance in the Github domain even without thresholding is notably higher than that in other domains, with the best method (zlib) achieving an AUC ROC of $\sim .70$. We speculate this is not because the GitHub domain, or code in general, is an easier domain to attack, but because the presumably reasonable decontamination threshold of $\leq$80% 13-gram overlap threshold only captures a small percentile of non-members as GitHub is naturally very high overlap.

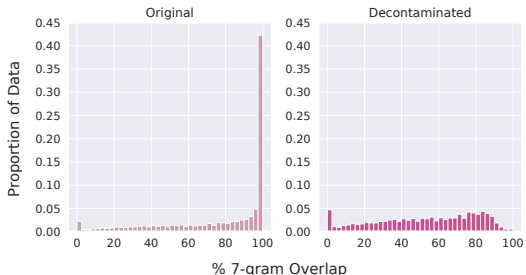

Figure 7: 7-gram overlap of GitHub non-member data before and after 13-gram decontamination at threshold $\leq$ 80%.

We speculate a large factor contributing to the high overlap is the repetitive nature of code, such as copyright notices, function definitions, and syntax like HTML tags. Figure 7 demonstrates how our decontamination threshold impacts the 7-gram distribution of non-members. Non-members under our decontamination threshold are more likely outliers to the GitHub domain (see Figure 15

for an example of such an outlier). The additional $n$-gram overlap threshold experiments (§3.2.2) only exacerbate the impact of thresholding, which leads to notably higher MIA performance.

Such observations indicate why lexical non-member boundaries may lead to ambiguous interpretations of MIA performance in high-overlap domains. Here, using semantic differences between samples to draw non-member boundaries may be key to better understanding membership leakage in such domains.

### B.4 Temporal Shift as Change in $n$-gram Overlap

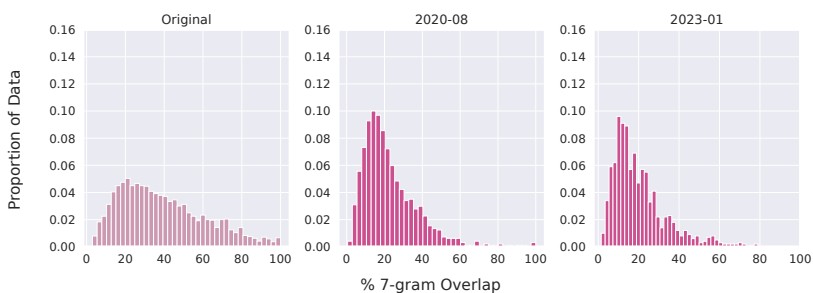

Figure 8: Distribution of $n$-gram overlap for non-member ArXiv preprints sampled from the months 2020-08 and 2023-06, respectively. We also plot the $n$-gram overlap distribution of the original Pile ArXiv non-members. Between the original non-members and both temporally shifted non-member sets, **the temporally shifted non-member $n$-gram overlap distributions are considerably more concentrated at lower % $n$-gram overlap**. The original non-members have an average 7-gram overlap of 39.3%, while non-members from months 2020-08 and 2023-06 have 7-gram overlap of 22.7% and 20.5%, respectively.

Figure 8 reinforces our observations in Figure 4, as similar to the temporal Wikipedia setting, temporally shifted non-members from after the target model's knowledge cutoff date are concentrated at considerably lower % $n$-gram overlap than non-members from the natural ArXiv non-member set. This contributes to the greater MIA performance in general over the temporally shifted ArXiv benchmarks. However, we note that $n$-gram overlap distribution shift does not provide a strong interpretation for the increase in MIA performance as non-members are increasingly temporally shifted. For example, the average 7-gram overlap of non-members from the month 2020-08 is 22.7% while the average for the month of 2023-06 is 20.5%. While there is a small decrease in average 7-gram for later non-members, the change is quite small and doesn't clearly justify the considerable difference in MIA performance when evaluating on benchmarks using non-members from the different months (i.e., .723 $\rightarrow$ .795 AUC ROC from the 2020-08 benchmark to the 2023-06 benchmark). We speculate other factors that contribute to this increase include changes in the distribution of topics (i.e., increasing popularity of research into LLMs) and the presence of specific identifying tokens (i.e., dates, references, new terminology). We believe such factors only further reinforce the need to carefully analyze MIA benchmark construction when evaluating MIAs to understand what signals are truly being captured.

## C   Characteristics of LLM Training

### C.1   Recency of Member Samples

We explore how the recency of member samples seen in training impacts MIA performance. We follow the same setup as the training data experiment (A.3.1), but instead of evaluating the checkpoint at step $n$ with the member data sampled from within steps $\{n - 100, n\}$ and the fixed non-member set, we fix the target model, only targeting the checkpoint at step 99000 for all the benchmarks.

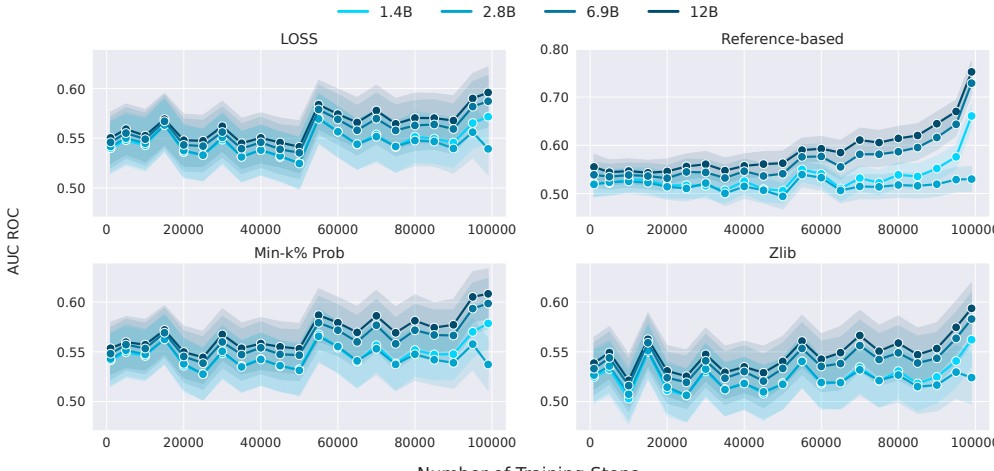

Figure 9: MIA performance for different member data sets sampled at different training steps across 1 epoch of the deduplicated Pile pretraining corpus, visualized across different attacks. Target model is the PYTHIA-DEDUP-12B checkpoint at step-99000. AUC-ROC reported. **Performance on benchmarks with more recently seen members is higher, but gradually decreases to a plateau for less recently seen members**.

Figure 9 demonstrates that, in general, member data seen more recently by the given checkpoint contributes to slightly higher MIA performance. We believe this supports existing work in LM forgetting (Jagielski et al., 2023), where observed patterns in recently seen training data are better preserved in the model parameters, while earlier seen data are less memorized.

We also note that the MIA performance trajectories seem to drop slightly more quickly for smaller models, though the trajectories across all model sizes seem to converge when evaluating on member data from much earlier in the training run. We speculate this is a result of larger models having more parameters, allowing them to capture more seen data before having to drop older knowledge.

We also note that, in the context of MIA against fine-tuning datasets, our results indicate that data seen during fine-tuning or continued pre-training may also be increasingly vulnerable due to how recent they are seen. This aligns with previous work demonstrating high MIA performance on fine-tuning datasets (Mireshghallah et al., 2022a; Fu et al., 2023). This is especially relevant in practice since fine-tuning is a popular option to re-purpose large pre-trained models for varying downstream tasks such as commercial use cases, which often involves tuning with sensitive data.

## C.2 Number of Training Epochs

While experiments against the DATABLATIONS model (Figure 2, right) operate in a fixed training data size setting, we also explore a more realistic setting where the amount of training data the target model sees increases alongside the effective epoch count by targeting the SILO-PDSW model and intermediate checkpoints. In Figure 10, for HackerNews, we observe MIA performance initially increases with more effective epochs, similar to the DATABLATIONS setting, but then begins to plateau or drop as effective epoch count continues to increase. DM mathematics, on the other hand, surprisingly decreases as the number of effective epochs increases. We speculate over factors that may contribute to these observations:

- HackerNews, even when up-sampled for this variant of the SILO-PDSW model, still only makes up 5.9% of the training data (Min et al., 2023). In the first few epochs, when the total training data seen is low, the model can memorize the HackerNews samples. However, as the number of epochs increases, the target model may tend to

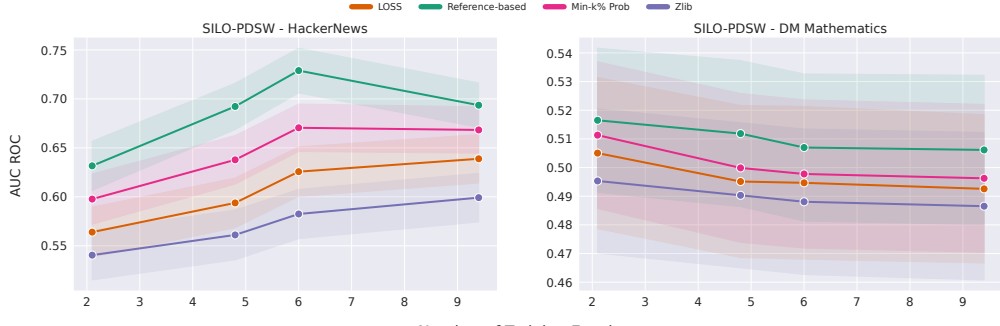

Figure 10: MIA performance on target model SILO-PDSW as the number of effective epochs in which the member domain data has been seen increases. AUC-ROC reported. For HackerNews, **performance does increase with an increasing number of effective epochs initially, but begins to plateau or even drop with further epochs**. For DM Mathematics, **performance surprisingly drops with increasing effective epochs**.

      overfit data more so from domains with greater representation. As the SILO model is on the smaller side with 1.4B parameters, we suspect the target model begins to memorize less of the HackerNews samples, leading to a plateau or drop in MIA performance.

- DM Mathematics also makes up only 3.5% of the training data. In addition, with DM Mathematics being a dataset of mathematical problems, we suspect that the abundance of tokens from a concentrated token space (i.e., digits, variables) that are largely symbolic rather than semantic makes memorization of specific samples unlikely. Overall, it simply fails to perform well on such data even after multiple epochs (as observed when looking at model loss values for this data).

For both cases, further investigation is needed into the target domains and attack setting setup to better understand these counter-intuitive phenomena.

## D  Revisiting Membership through Semantic Similarity

We follow the same pipeline as §5 but focus on generating member paraphrases that are semantically similar while preserving as much specific information from the original member sample as possible. To do so, we follow Yang et al. (2023) and prompt GPT4 (OpenAI et al., 2024) to paraphrase member samples in a different style (5 trials per member). We perform this paraphrasing over the ArXiv, Wikipedia, and HackerNews domains; see Table 9 for domain-specific prompts. In general, we don't focus on the lexical similarity of the paraphrases unlike the earlier semantically similar samples generated via masking and replacing a small percentage of tokens. However, with HackerNews, we do specify a comment structure different from how HackerNews records were formatted for model training, a noticeable lexical difference.

| Domain | Prompt |
|---|---|
| ArXiv | Please help me paraphrase the following text chunk from a research paper in a different style.  Importantly, for sentences containing specific details like mathematical definitions or proofs, only make minimal changes and ensure these details are included exactly in the paraphrase. If the paper includes a title or authors, please keep them in the rephrase. If not, please DO NOT make up a title. Use a similar number of words. |
| Wikipedia | Please help me paraphrase the following text chunk from Wikipedia in a different but concise style. Importantly, for sentences containing specific details, make minimal changes and ensure all details are included correctly in the paraphrase. Use a similar number of words. |
| HackerNews | Please help me paraphrase the following conversation chunk from a thread in HackerNews while maintaining the conversational style. Follow this structure for each comment in the thread: [user] - [comment].  Ensure all user's comments are represented in the paraphrase. Make sure all details in each user's comments are included correctly in the paraphrase, such as links.  Be specific and don't generalize. |

Table 9: Instructions used to prompt GPT4 to obtain paraphrased members.

We again visualize the score distributions between the paraphrased members, and original members and non-members (Figure 11). We observe in general across both the LOSS and Reference-based attacks over the three domains that the paraphrased member score distributions are distinguishable from the original member and non-member score distributions but have noticeable overlap, similar to what was observed with masking-based semantic neighbors. However, when we perform the FPR experiment (Table 10), we see that in high-confidence settings, the paraphrased members are likely to be classified as non-members. Both the LOSS and Reference-based attacks seem noticeably insensitive to such paraphrased neighbors.

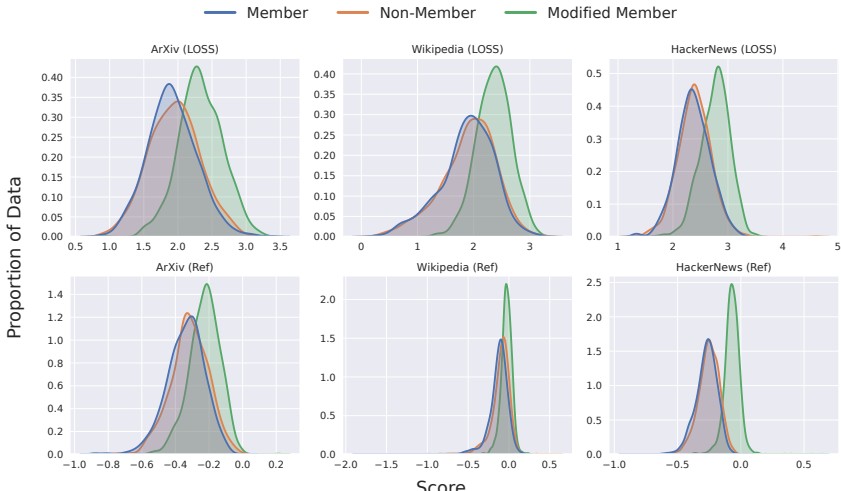

Figure 11: Distribution of scores for LOSS and Reference-based attacks for members, non-members, and GPT4-paraphrased (modified) members across ArXiv, Wikipedia, and HackerNews domains.

| Domain | LOSS | | | Ref | | |
|---|---|---|---|---|---|---|
| | 1% | 5% | 10% | 1% | 5% | 10% |
| ArXiv | 0.1 | 0.2 | 0.5 | 0.2 | 0.7 | 1.7 |
| Wikipedia | 0.0 | 0.0 | 0.2 | 0.0 | 0.2 | 0.7 |
| HackerNews | 0.1 | 1.1 | 1.7 | 0.0 | 0.1 | 0.2 |

Table 10: FPR (%) on modified members (treated as non-members) when using a score threshold that achieves a 1, 5, or 10% FPR on the original member and non-member data for the ArXiv, Wikipedia, and HackerNews domains. Modified members are generated by prompting GPT4 to paraphrase member samples with significant lexical difference. LOSS and Reference-based attack reported.

# E Additional Figures and Tables

| # Params | Wikipedia | | | | | Github | | | | | Pile CC | | | | | Pubmed Central | | | | |
|---|---|---|---|---|---|---|---|---|---|---|---|---|---|---|---|---|---|---|---|---|
| | LOSS | Ref | min-$k$ | zlib | Ne | LOSS | Ref | min-$k$ | zlib | Ne | LOSS | Ref | min-$k$ | zlib | Ne | LOSS | Ref | min-$k$ | zlib | Ne |
| 70M | 0.8 | 0.9 | **1.0** | **1.0** | 0.5 | **13.1** | 7.8 | 12.6 | 13.0 | 11.0 | 0.7 | **1.0** | 0.4 | 0.4 | 0.1 | 0.7 | **1.0** | 0.9 | 0.5 | 0.6 |
| 160M | 1.1 | 0.8 | 1.2 | **1.4** | 1.3 | 13.5 | 4.6 | 12.3 | **14.7** | 5.9 | 0.4 | **0.8** | 0.5 | 0.4 | 0.4 | 0.7 | 0.9 | **1.0** | 0.3 | 0.1 |
| 1.4B | 0.6 | **0.9** | 0.5 | 0.7 | 0.4 | 12.8 | 0.7 | 12.9 | **16.4** | 3.9 | 0.6 | 0.6 | 0.5 | 0.7 | **0.8** | 0.4 | **0.7** | 0.6 | 0.5 | 0.1 |
| 2.8B | 0.6 | 0.8 | 0.5 | 0.7 | **0.9** | 20.8 | 4.5 | 20.8 | **23.4** | 11.1 | 0.6 | 0.5 | 0.7 | 0.8 | **0.9** | 0.4 | 1.0 | **1.4** | 0.6 | 0.9 |
| 6.9B | **0.6** | **0.6** | 0.4 | **0.6** | 0.5 | 12.9 | 0.6 | 13.1 | **16.8** | 6.1 | 1.0 | **1.4** | 1.2 | 1.3 | 1.0 | 0.8 | **1.6** | 0.8 | 0.3 | 0.8 |
| 12B | **0.7** | 0.6 | 0.6 | **0.7** | 1.0 | 13.9 | 0.8 | 14.2 | **17.4** | 4.9 | 1.0 | **1.7** | 1.1 | 1.5 | 1.0 | 1.0 | **1.5** | 1.3 | 0.7 | 0.9 |

| # Params | ArXiv | | | | | DM Math | | | | | HackerNews | | | | | The Pile | | | | |
|---|---|---|---|---|---|---|---|---|---|---|---|---|---|---|---|---|---|---|---|---|
| | LOSS | Ref | min-$k$ | zlib | Ne | LOSS | Ref | min-$k$ | zlib | Ne | LOSS | Ref | min-$k$ | zlib | Ne | LOSS | Ref | min-$k$ | zlib | Ne |
| 70M | 0.8 | **1.0** | 0.7 | 0.5 | 0.9 | 0.6 | **1.3** | 0.5 | 0.7 | 0.8 | 1.1 | 0.7 | **1.3** | **1.3** | 0.5 | **2.2** | 1.4 | 1.8 | 2.1 | 2.0 |
| 160M | **0.8** | 0.4 | 0.2 | 0.7 | 0.3 | 0.5 | **1.4** | 0.6 | 1.2 | 0.7 | 1.0 | 0.8 | **1.2** | 0.6 | 0.7 | **2.4** | 1.3 | 2.0 | 2.2 | 2.2 |
| 1.4B | 0.3 | **1.0** | 0.2 | 0.4 | 0.7 | 0.8 | 0.8 | 0.6 | 1.0 | **1.7** | 0.7 | 0.9 | **1.2** | 0.9 | 0.8 | **2.4** | 1.4 | **2.4** | 2.3 | 2.3 |
| 2.8B | 0.5 | **2.1** | 0.5 | 0.5 | 0.5 | 0.8 | 0.4 | 1.0 | **1.3** | 0.8 | 0.6 | 1.4 | 0.8 | 1.1 | **1.7** | **2.8** | 2.2 | **2.8** | **2.8** | 2.4 |
| 6.9B | 0.6 | **1.8** | 0.6 | 0.6 | 0.6 | 0.9 | 0.2 | 0.6 | **1.0** | 0.7 | .9 | **1.9** | 1.0 | 0.9 | 1.3 | **2.6** | 1.8 | 2.5 | 2.5 | 2.2 |
| 12B | 0.6 | **2.5** | 0.6 | 0.5 | 0.9 | 1.0 | 0.5 | 0.5 | **0.9** | 0.8 | 0.7 | **2.3** | 0.8 | 0.8 | 1.4 | **2.7** | 1.8 | 2.6 | 2.6 | 2.2 |

Table 11: %TPR@1%FPR of MIAs against Pythia-dedup across different datasets from the Pile. The highest performance across the different MIAs is bolded per domain. In general, **leakage in high confidence settings is low ($< 3\%$)**. As with AUC ROC, GitHub is an exception, still yielding considerably higher leakage with most attacks. Unlike with AUC ROC, trends in performance are much noisier in the high-confidence setting, with trends in model size and best-performing attacks in certain domains no longer holding, reinforcing the difficulty in determining a *best* attack.

| # Params | Wikipedia | | | | Github | | | | Pile CC | | | | Pubmed Central | | | |
|---|---|---|---|---|---|---|---|---|---|---|---|---|---|---|---|---|
| | LOSS | Ref | min-$k$ | zlib | LOSS | Ref | min-$k$ | zlib | LOSS | Ref | min-$k$ | zlib | LOSS | Ref | min-$k$ | zlib |
| 160M | .503 | **.512** | .491 | **.512** | .657 | .639 | .652 | **.674** | .496 | .491 | **.504** | .497 | .499 | **.513** | .506 | .500 |
| 1.4B | .513 | **.552** | .511 | .520 | .698 | .670 | .699 | **.710** | .501 | **.522** | .510 | .502 | .498 | **.531** | .502 | .500 |
| 2.8B | .518 | **.582** | .518 | .525 | .712 | .653 | .713 | **.723** | .501 | **.537** | .508 | .504 | .500 | **.537** | .504 | .501 |
| 6.9B | .528 | **.618** | .536 | .536 | .730 | .644 | .733 | **.739** | .507 | **.550** | .515 | .509 | .506 | **.558** | .511 | .506 |
| 12B | .535 | **.639** | .544 | .544 | .740 | .630 | .743 | **.748** | .511 | **.567** | .517 | .512 | .513 | **.582** | .523 | .512 |

| # Params | ArXiv | | | | DM Math | | | | HackerNews | | | | The Pile | | | |
|---|---|---|---|---|---|---|---|---|---|---|---|---|---|---|---|---|
| | LOSS | Ref | min-$k$ | zlib | LOSS | Ref | min-$k$ | zlib | LOSS | Ref | min-$k$ | zlib | LOSS | Ref | min-$k$ | zlib |
| 160M | **.510** | .494 | .507 | .502 | .489 | **.510** | .495 | .481 | .494 | .491 | **.509** | .498 | .503 | **.511** | .505 | .506 |
| 1.4B | .515 | .516 | **.517** | .509 | .486 | **.512** | .497 | .482 | .505 | **.522** | .512 | .504 | .505 | **.522** | .510 | .508 |
| 2.8B | .519 | **.531** | .525 | .514 | .484 | **.505** | . 480 | .480 | .513 | **.551** | .524 | .509 | .508 | **.533** | .513 | .511 |
| 6.9B | .529 | **.558** | .535 | .523 | .485 | **.511** | .496 | .481 | .521 | **.579** | .536 | .513 | .514 | **.554** | .522 | .516 |
| 12B | .534 | **.575** | .546 | .527 | .485 | **.510** | .497 | .481 | .528 | **.606** | .546 | .517 | .519 | **.569** | .528 | .520 |

Table 12: AUC ROC of MIAs against non-deduped Pythia across different datasets from the Pile. The reference-based attack uses STABLELM-BASE-ALPHA-3B-V2 as the reference model. The highest performance across the different MIAs is bolded per domain. **Performance follows similar trends seen when targeting the Pythia-dedup models, but performance is, in general, marginally higher**. Due to computational limitations, we leave out evaluations for the Neighborhood attack, but expect similar trends.

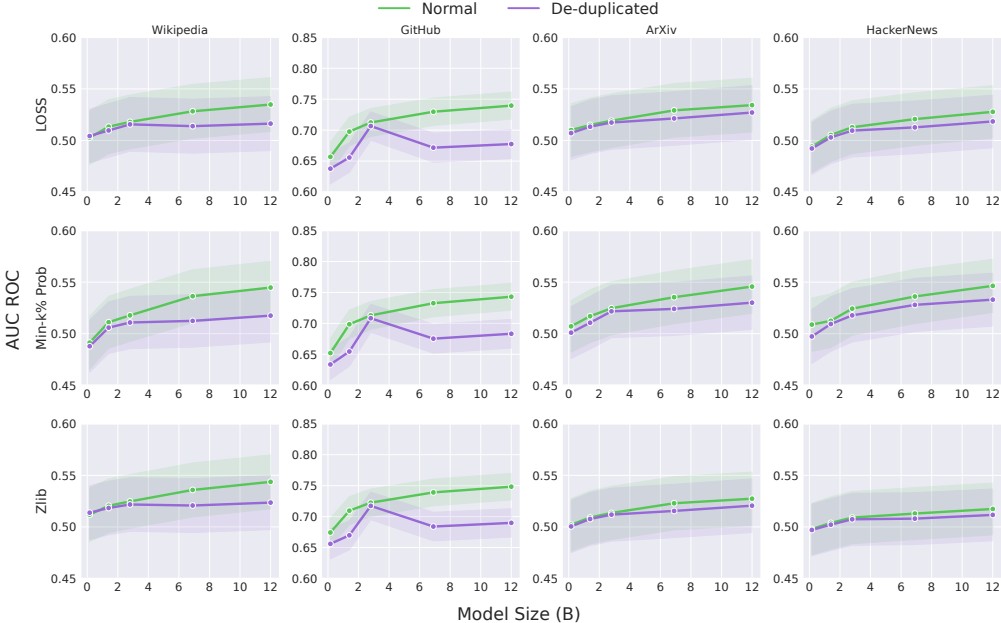

Figure 12: MIA performance as model size increases over select domains for various other attacks. We additionally plot the AUC ROC trajectory against the non-deduped Pythia suite for comparison. Similar to the reference-based attack, **increasing model size slightly boosts MIA performance while deduplication decreases performance**.

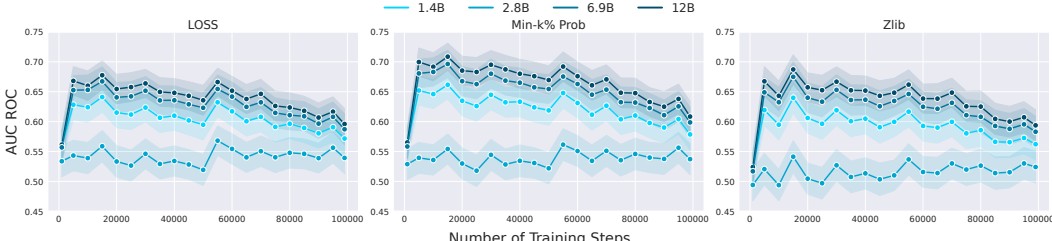

Figure 13: MIA performance as the amount of training data seen increases across 1 epoch of the deduplicated Pile pretraining corpus, visualized over a range of model sizes for various attacks. We use the training step as a unit for the amount of training data seen, with 1 step corresponding to seeing 2097152 tokens. AUC-ROC reported. Similar to the reference-based attack, for all attacks, **performance drastically increases before gradually decreasing as the amount of training data seen increases**.

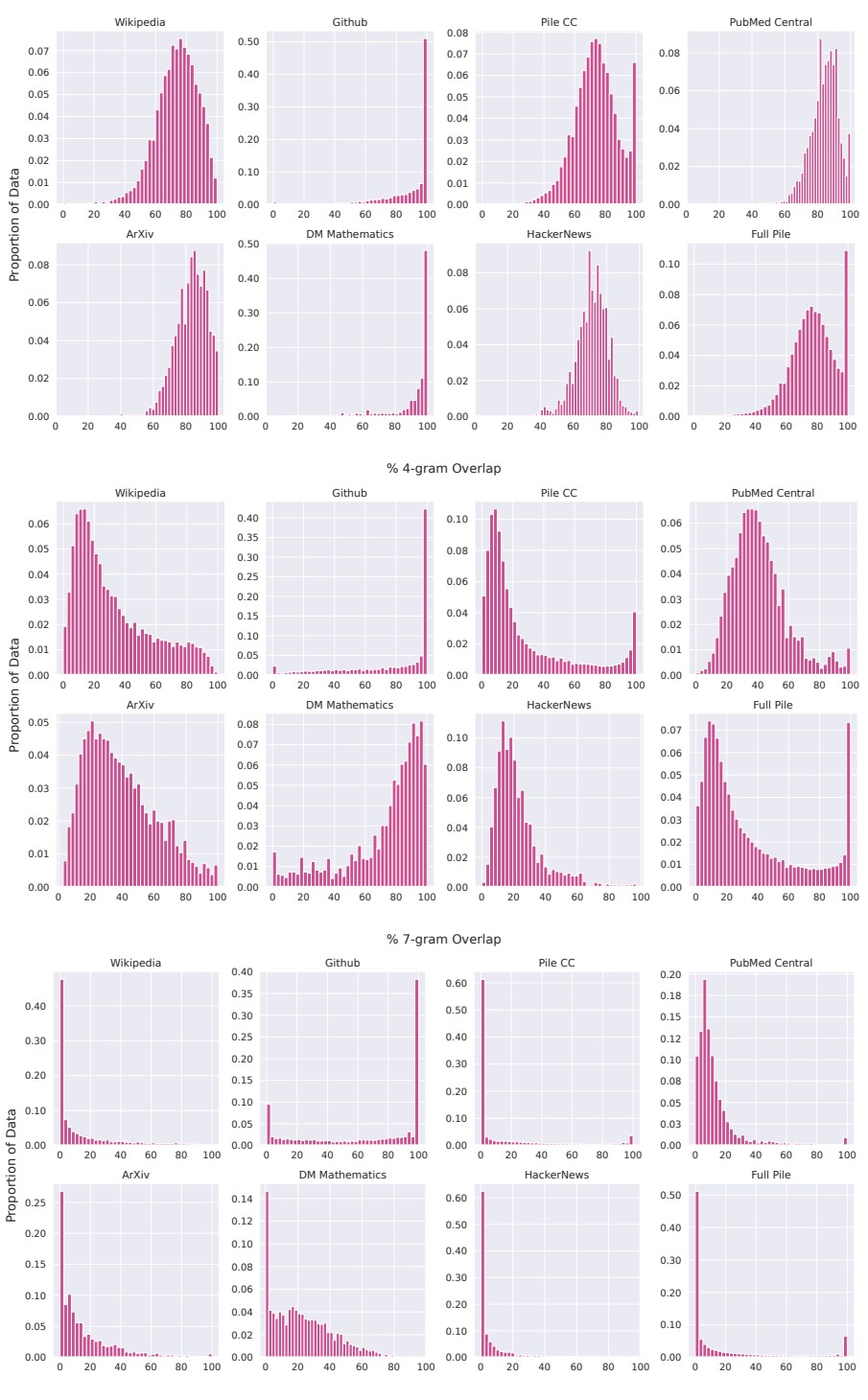

Figure 14: Distribution of $n$-gram overlap over all evaluation domains for $n = 4, 7, 13$.

http://burmese.voanews.com/a/myanmar-ambassador-of-thailand-said-they-will-appeal-the-case-according-to-the-thai-law/3124176.html

လိပ္ကၽြန္းအမႈ အယူခံ၀င္ဖို႔ ျပင္ဆင္

ၿဗိတိန္ႏိုင္ငံသား ကမာၻလွည့္ခရီးသည္ ႏွစ္ဦးကို ထိုင္းႏိုင္ငံ ေကာ့ေတာင္ အပန္းေျဖကၽြန္းမွာ သတ္ခဲ့တယ္ဆိုတဲ့စြဲခ်က္နဲ႔ ေသဒဏ္ ခ်ခံထားရတဲ့ ျမန္မာႏိုင္ငံသားႏွစ္ဦးရဲ႕အမႈကို ထိုင္းႏိုင္ငံရဲ႕ တရားဥပေဒအတိုင္း ဆက္ၿပီး အယူခံ၀င္သြားႏိုင္ဖို႔အတြက္ ထုိင္းႏိုင္ငံ ေရွ႕ေနမ်ားေကာင္စီနဲ႔ ဘန္ေကာက္ၿမိဳ႕က ျမန္မာသံအမတ္ပါ၀င္တဲ့ လိပ္ကၽြန္းအမႈအတြက္ လိုက္ပါေဆာင္ရြက္ေနတဲ့ ျမန္မာသံ႐ုံးအထူးေကာ္မတီတို႔ ေတြ႔ဆံုခဲ့ၿပီး သက္ေသအေထာက္အထား အခ်က္အလက္ စုေဆာင္းဖို႔ အတြက္ျပင္ဆင္ေနၾကပါတယ္။ မေအးေအးမာကသတင္းေပးပို႔ထားပါတယ္ ။

ၿဗိတိန္ႏိုင္ငံသား ကမာၻလွည့္ခရီးသည္ ႏွစ္ဦးကို ထိုင္းႏိုင္ငံ ေကာ့ေတာင္ အပန္းေျဖကၽြန္းမွာ သတ္ခဲ့တယ္ဆိုတဲ့ စြဲခ်က္နဲ႔ ေသဒဏ္ ခ်ခံထားရတဲ့ ျမန္မာႏိုင္ငံသား ႏွစ္ဦးရဲ႕ အမႈနဲ႔ ပတ္သက္လို႔ လိပ္ကၽြန္းအမႈအတြက္ ဖြဲ႔စည္းထားတဲ့ ျမန္မာသံ႐ုံးအထူးေကာ္မတီ၀င္ေတြ နဲ႔ ထုိင္းေရွ႕ေနမ်ားေကာင္စီနဲ႔ တိုင္ပင္ေဆြးေႏြးမႈေတြလုပ္ခဲ့ အၿပီးမွာေတာ့ ျမန္မာသံအမတ္ဦး၀င္းေမာင္က ထုိင္းႏိုင္ငံရဲ႕ တရားဥပေဒအတုိင္းဆက္ၿပီး အယူခံ၀င္သြားမယ္လုိ႔ ေျပာဆုိသြားပါတယ္။

"ထိုင္းႏိုင္ငံအစိုးရနဲ႔ ထိုင္းႏိုင္ငံရဲ႕ ဥပေဒအရ ထိုင္း၀န္ႀကီးခ်ဳပ္ကိုယ္တိုင္ အယူခံ၀င္ႏိုင္ေၾကာင္း လမ္းခင္းေပးတဲ့အတြက္ က်ေနာ္တို႔ ဆက္ၿပီး အယူခံ၀င္ၿပီးေတာ့ ဒီကေလးေတြရဲ႕ေမွ်ာ္လင့္ခ်က္ကို ဆက္လက္ၿပီး ေဆာင္ရြက္သြားမွာျဖစ္ပါတယ္။"

ထိုင္းေရွ႕ေနမ်ားေကာင္စီဟာ ေရွ႕ေန ၁၁ ဦးပါအဖြဲ႕နဲ႔ျမန္မာလူငယ္ ၂ ဦးရဲ႕အမႈကုိ အခမဲ့ ေရွ႕ေန လိုက္ေပးခဲ့ၿပီး တရားမွ်တစြာ စီရင္ႏိုင္ဖို႔အတြက္ ဥပေဒေၾကာင္းအရ အကူညီေပးေနခဲ့တာပါ။ သူတို႔အေနနဲ႔ ခိုင္မာတဲ့ သက္ေသ အေထာက္အထား ေတြလံုေလာက္စြာမရခဲ့တာေၾကာင့္ ဒီအမႈကို လုိက္ပါေဆာင္ရြက္တဲ့အခါမွာ လိုအပ္ခ်က္ေတြလည္းရွိေနခဲ့တယ္လို႔ ဆိုပါတယ္ ။ ဒီကေန႔မွာေတာ့ ထိုင္းေရွ႕ေနမ်ားေကာင္စီက ျမန္မာသံအမတ္နဲ႔ လိပ္ကၽြန္း အတြက္ ဖြဲ႔စည္းထားတဲ့ ျမန္မာသံ႐ုံးအထူးေကာ္မတီနဲ႔ ေတြ႔ဆံုၿပီး ျမန္မာလူငယ္ ၂ ဦး အတြက္ အယူခံဆက္တက္ႏိုင္ဖို႔ဥပေဒေၾကာင္းအရ ဆက္လက္ လုပ္ေဆာင္ႏိုင္တဲ့ အခ်က္ေတြကုိ ေဆြးေႏြးေျပာဆိုခဲ့တယ္လို႔ သံ႐ုံးအထူးေကာ္မတီေခါင္းေဆာင္ျဖစ္တဲ့ ဦးေက်ာ္ေသာင္းက ေျပာပါတယ္။

"ေရွ႕ေနမ်ားေကာင္စီ ဥကၠ႒က Dej Udom Krairit ေပါ့ေနာ္။ သူကေျပာတာက အခ်က္ ၃ ခ်က္ေျပာတယ္။ အယူခံကိစၥမွာတဲ့ အမႈကို အစကေနအဆံုးအထိ ျပန္ၿပီးေတာ့ စစ္ေဆးေပးဖို႔ ေတာင္းဆိုႏိုင္စရာကိစၥရွိတယ္တဲ့။ အဲဒါေတြကေတာ့ ဘာလဲဆိုေတာ့ နံပတ္ ၁ သက္ေသခံေတြဟာ လြဲမွားတယ္။ သက္ေသေတြလည္းလြဲမွားတယ္ဆိုၿပီးေတာ့ က်ေနာ္တို႔ဘက္က တရားခံဘက္က ေခ်ပႏိုင္တဲ့အခ်က္အလက္ရွိရင္ တင္ျပလို႔ရတယ္တဲ့။ နံပတ္ ၂ အခ်က္ကေတာ့ ပစၥည္းသက္ေသေတြဟာ အတုအေယာင္ေတြျဖစ္ေနတယ္။ အစစ္အမွန္မဟုတ္ဘူး။ ေနာက္ၿပီးေတာ့ တင္ျပတဲ့အခ်က္အလက္ေတြကလည္း အမွားအယြင္းေတြျဖစ္ေနတယ္ဆိုလို႔ရွိရင္လည္း ျပန္ၿပီးေတာ့မွ အမႈကို အစအဆံုးျပန္ၿပီးေတာ့ စစ္ေဆး လို႔ရပါတယ္။ နံပတ္ ၃ အခ်က္က ထပ္ၿပီးေတာ့ တင္ျပႏိုင္တဲ့ သက္ေသေတြ ပစၥည္းေတြ၊ သက္ေသက သက္မဲ့၊သက္ရွိ သက္ေသေတြ ရွိခဲ့ရင္ ဒီအခ်က္ ၃ ခ်က္ထဲက တခ်က္ခ်က္ ကိုယ့္ဘက္က ေထာက္ျပႏိုင္စရာ ေျပာႏိုင္စရာရွိရင္ ဒီအမႈကို အစအဆံုးထိျပန္ၿပီးေတာ့ စစ္ေဆးခြင့္တင္လို႔ရပါတယ္တဲ့။ ဒီအခ်က္ ၃ ခ်က္ထဲမွာ တခုခုေတာ့လည္းေကာင္း ၊ အခ်က္ ၂ ခ်က္ေသာ္လည္းေကာင္း၊ အခ်က္ ၃ ခ်က္ေသာ္လည္းေကာင္း ၊ တခုခုရွိခဲ့လို႔ရွိရင္ မိသားစု၀င္တဦးဦးက ျပန္ၿပီးေတာ့ ဒီအမႈကိစၥကို ျပန္လည္စစ္ေဆးေပးဖို႔အတြက္ ေတာင္းဆိုလို႔ရပါတယ္တဲ့။"

ရန္ကုန္အပါအ၀င္ နယ္ၿမိဳ႕ေတြမွာေတာ့ ျမန္မာႏိုင္ငံသားလူငယ္ ၂ ဦးကိုျပန္လႊတ္ေပးဖို႔နဲ႔ တရားမွ်တမႈကို ေတာင္းဆိုတဲ့ ဆိုင္းဘုတ္ေတြကို ကိုင္ေဆာင္ျပဆႏၵျပပဲြေတြက ျဖစ္ေနဆဲပါ။

ဆႏၵျပသူေတြဟာ ထုိင္းႏိုင္ငံရဲ႕ တရားစီရင္ေရးကုိလည္း ေ၀ဖန္ၾကတာေတြလည္းရွိေနပါတယ္။ ဒီအတြက္ ထုိင္းေရွ႕ေနမ်ားေကာင္စီက ထိုင္းႏိုင္ငံရဲ႕ တရားစီရင္ေရး မ႑ိဳင္ဟာ တည့္မတ္ မွန္ကန္တာကို ျပသဖုိ႔အတြက္ ဒီအမႈနဲ႕ပတ္သက္လို႔ ခိုင္မာတဲ့ သက္ေသအေထာက္အထားေတြေပးဖို႔ ေတာင္းဆိုခဲ့တယ္လုိ ့ဦးေက်ာ္ေသာင္းက ဆက္ေျပာပါတယ္။

"ငါတို႔တိုင္းျပည္ရဲ႕တရားစီရင္ေရးမ႑ိဳင္ ဘယ္ေလာက္ တည့္မတ္ မွန္ကန္သလဲဆိုတာျပခ်င္တယ္။ နင္တို႔ေတြ ဒီကေနၿပီးေတာ့ ဒီလိုခိုင္မာေသခ်ာတဲ့ သက္ေသအခ်က္အလက္ေတြ ထြက္လာလိမ့္မယ္လို႔ ငါတို႔ေမွ်ာ္လင့္တယ္။ အျမန္ဆံုး ငါတို႔ကို ကူညီေပးပါလို႔ ေမတၱာရပ္ခံပါတယ္။

ၿပီးခဲ့တဲ့ တႏွစ္တာကာလအတြင္းမွာေတာ့ သက္ေသထြက္ေပးႏုိင္တဲ့ သူေတြရဲ႕ ေနထုိင္ေရးအလုပ္အကိုင္နဲ႔ လံုျခံဳေရးကိစၥေတြကို ျမန္မာသံ႐ုံးနဲ႔ လိပ္ကၽြန္းအထူးေကာ္မတီတို႔က တာ၀န္ယူေပးႏိုင္တာေတြ မရွိတဲ့အတြက္ လိပ္ကၽြန္းေပၚမွာ အလုပ္လုပ္ၾကတဲ့ ျမန္မာႏိုင္ငံသားသက္ေသေတြ ရသင့္သေလာက္ မရရွိခဲ့ဘူးလို႔ ဦးေက်ာ္ေသာင္းကဆုိပါတယ္။

ဒီကေနမွာေတာ့ရန္ကုန္အပါအ၀င္ ထိုင္း-ျမန္မာနယ္စပ္ၿမိဳ႕ေတြျဖစ္တဲ့ ျမ၀တီၿမိဳ႕နဲ႔ ဘုရားသံုးဆူၿမိဳ႕ ေတြမွာေတာ့ ဆႏၵျပမႈေတြ ဆက္ျဖစ္ေပၚေနပါတယ္။

Figure 15: A sample GitHub non-member outlier captured by the $\leq$ 80% 13-gram overlap threshold. This sample is from a language resource repository under Google, but is a clear outlier to the code-dominant GitHub domain

