# OpenReview forum: "Do Membership Inference Attacks Work on Large Language Models?"
_colmweb.org/COLM/2024/Conference — COLM_

### Official Review · Reviewer_Rszs · 2024-04-17

**Rating:** 7
**Confidence:** 3
**Ethics Flag:** 1

**Summary:**

This paper presents an empirical study of the Membership Inference Attack (MIA) against pre-trained large language models (LLMs). Following comprehensive experiments, the authors discovered that current membership inference methods face challenges when applied to LLMs. Further analysis attributed this poor MI performance to several factors: (1)The nature of pre-training, which involves a large volume of training data, typically trained for only a single epoch.(2)The inherent text-repeating nature of natural language, which introduces difficulties into membership inference. Additionally, the authors discussed the importance of addressing distributional shifts and suggested extending the MI game to consider neighboring members to better interpret information leakage.

**Questions To Authors:**

(1)In Section 4, the modified member is derived from member data, so it should be more similar to member data compared to non-members, or it should be harder to distinguish (because modified member should have a greater n-gram overlap). Therefore, intuitively, the MI score of modified members should fall between non-members and members. However, the results in Figure 5 are far from this expectation. What could be the reason for this?

(2)Figure 7 illustrates the 7-gram overlap of the GitHub domain before and after decontamination. This suggests that the statistics presented in Figure 3 seem to rely on data from the original domain. However, this approach lacks rigor since your main results are derived from decontaminated data, making it more appropriate to conduct statistical analysis based on decontaminated data. Please correct me if I have overlooked any details or misunderstood anything.

**Reasons To Accept:**

(1) The paper is well structured and logically flows smoothly.

(2) The paper is sound. The authors verify their findings and hypotheses through sufficient experiments and analysis.

**Reasons To Reject:**

(1) The paper is somewhat dense for an 9-page document. It encompasses various topics, ranging from the findings regarding the challenges of MI in LLMs and analysis of the reasons behind them to discussions on unintended distribution shifts and the MI game itself. This density can make it a bit challenging to follow and grasp the details, especially without referencing the appendix.

(2) A small suggestion instead of a reason for rejection: It would be better to place Table 1 on page 3 or 4 for smoother reading. Additionally, Figure 5 is currently somewhat complex and unclear for readers, especially because the bar graphs for non-members and members largely overlap, leading to confusion upon initial glance.

(3) I have some doubts about the empirical results, please refer to the Question part below

---

> ### Author Rebuttal · Authors · 2024-05-31
>
> We thank the reviewer for their helpful feedback and for supporting our paper.
>
> *[Reasons to reject]* Table/Figure clarification
>
> We will implement the suggestion regarding Table 1 in our revision. For Figure 5, we wanted to show how the original member/non-member score distributions highly overlap compared to that of the modified members. One clarification we will try is replacing the histograms with KDE curves.
>
> *[Q1]* MI score distribution of modified members
>
> The reviewer is right in their reasoning: for a modification heuristic that preserves language structure perfectly and does not make a lot of edits, this would indeed be the case. However, the caveat here lies in “how” these modified records are generated.
>
> For members derived within some edit-distance, the token replacements are random. For instance, “Dug is a good dog” could become “Dug is cactus good dog” for n=1. While this is indeed closer to the member record than  some non-member “Gato is a good cat”, the unnatural language structure from random tokens can confuse the model (higher perplexity) than a naturally-structured non-member would, especially for models that generalize well. A reference-based MIA is impacted by the phenomenon, where the reference model itself would also exhibit similar behavior in its loss, though perhaps not as significantly.
>
> This disparity is less for semantically-similar members for both MIAs, likely due to the BERT token replacements preserving some of the semantics. However, as the BERT model is a weaker model, the replacement tokens under BERT may be noticeably unlikely under the target model.
>
>
> *[Q2]* Visualization of n-gram overlap distributions
>
> The reviewer’s understanding is correct that the main results are derived from a benchmark where non-members undergo a soft decontamination. We chose to use the unfiltered data to visualize n-gram overlap distributions primarily to demonstrate the considerably high overlap between natural member/non-member data. We understand, however, that this could be confusing, especially for domains such as GitHub, where the naturally high overlap conflicts with the high AUCs against the (decontaminated) GitHub data (current analysis in Appendix B.3). We will add n-gram overlap distributions over the decontaminated non-member sets as well for other domains as we did with GitHub.

---

> > ### Comment · Reviewer_Rszs · 2024-06-05
> > **Response to authors**
> >
> > Thanks for your response. I'd like to raise my score to 7 and remain positive about the acceptance of this paper.

---

### Official Review · Reviewer_sAnK · 2024-04-28

**Rating:** 6
**Confidence:** 5
**Ethics Flag:** 1

**Summary:**

This paper questions the effectiveness of MIA approaches which are currently used to evaluate the leakage of large langage model (LLM) pre-training data. The authors evaluate several MIA methods from previous works. They find that the performance of MIAs is quite poor on several LLMs trained on the Pile when evaluated on subsets of the Pile from different domains. This finding seems to contradict previous works which reported much better MIA performance. The authors explain this difference by arguing that the previous works used member and non-member data from different time ranges, resulting in distribution shift of non-members with respect to members. Therefore, previous MIAs likely performed distribution inference rather than membership inference, with distribution inference being an easier task. The authors also question the relevance of the task’s definition for natural lanauage where it is possible to construct different but similar samples (which are technically non-members), but should reasonably be viewed as members. MIAs are shown not to be robust to such samples, which are wrongly classified as non-members in the high-confidence regime.

I think that this work is timely and tackles an important question, which is relevant for privacy auditing and for copyright infringment. The paper does not propose a new method, so the technical novelty is very limited. However, it performs a large-scale evaluation of several state-of-the-art MIA approaches, whose results challenge the strong performances of MIAs reported in some of the previous works (Shi et al, 2023 and Meeus et al, 2023). This is an intriguing finding and the authors propose a plausible explanation for these results. The authors further challenge the definition of MIAs in the context of LLMs, showing that MIAs are not robust to small changes in the sample.

However, the paper in its current form does not seem ready for publication, as it is imprecise on several important points neccessary to substantiate its claims and the benchmarking of MIAs seems incomplete and to suffer from a methodological flaw. Most of these points seem fixable and I hope that the authors will address them in their rebuttal.

**Questions To Authors:**

- How do the authors ensure that their MIA evaluation setting (non-members from the Pile test data) does not suffer from distribution shift ? For instance, is there any reference stating that the Pile data was randomly shuffled before being split into train and test ?

- Other works have already challenged the reliability of MIA predictions, see [A] and [B]. For instance, [A] shows that neighboring non-member samples would be classified as members which is related to the point made in this paper. Therefore the insight that MIAs are brittle is not new (although demonstrating it empirically in the LLM setting is). The paper does not position itself with respect to these works. Can the authors elaborate on their differences with [A] and [B] and add a discussion of previous works challenging MIA reliability ?

[A] Rezaei, S., & Liu, X. (2022). On the Discredibility of Membership Inference Attacks. arXiv preprint arXiv:2212.02701.

[B] Hintersdorf, D., Struppek, L., & Kersting, K. (2021). To trust or not to trust prediction scores for membership inference attacks. arXiv preprint arXiv:2111.09076.

- How does the state-of-the-art MIA method of Meeus et al, 2023 perform compared to the other approaches?

- Figure 1: How are the confidence intervals computed, e.g.., over how many samples?

- Sec 3.2.2 – Resampling of non-members to have low overlap. This step seems over-engineered and, as the authors notice, results in distribution shift. A simpler approach would be to rank the non-members by the MIA scores and check if MIA confidence is negatively correlated with n-gram overlap. Did the authors consider this approach ?

- Benchmarking of MIA approaches : Are the results statistically significant ?

- ArXiv temporal shift experiment: there is a difference in the pipeline used to construct members and non-members (S3 vs arXiv API), which could introduce a distribution shift beyond the temporal shift. To substantiate the claim that « both arXiv bulk access and API should yield the preprints in the same manner regardless », can the author apply the API to retrieve some of the data before July 2020 and check that after pre-processing the documents match ?

Other comments:

- Please consider adding the definition of a sample (200-length documents) in the main paper.

- Appendix A.4: I disagree with the statement that the reference-based attack is largely similar to the offline LiRA attack of Carlini et al., 2022. Two key features of LiRA, which don't feature in the reference-based attack, are its use of the logit loss (log(p/(1-p)) and fitting a Gaussian distribution to its distribution, which allows extrapolating to the high-confidence regime. The reference-based attack would be more similar if the log(1-p) terms were substracted from the loss.

- Reference-based attack: The authors acknowledge the limitation of LLM reference models: although they are not supposed to be trained on the target sample (for the one-sided hypothesis test to be valid), they likely are. I agree with the authors that this seems hard to fix. However, because the attack’s assumptions are invalid, it’s hard to interpret why and when it works. Some analysis is provided in Appendix B.2; however it would be better to explain the first time the attack is presented (in the main paper) when the attack is likely to work in spite of the invalid assumptions and why. Perhaps this intuition can also guide the choice of the reference model.

Minor :
- First paragraph of Sec. 3.2.1 – the argument needs to be completed by explaining that MIA success if linked to overfitted (Yeom et al, 2018).

- Figure 2 : the nuances of blue on the left are hard to distinguish in print, please consider increasing the contrast and the font size.

- n-gram overlap formula: Please consider adding 1-2 sentences explaining what the formula is meant to capture.

- Figure 3 – Github: is my understanding correct that on Github 40 % of the samples have 100 % n-gram overlap with the training dataset ? Please consider discussing this in the main paper and explaining that this does not mean that these non-members are members.

- Please consider fully defining the loss in the main paper using the negative log-likelihood of tokens.

**Reasons To Accept:**

- The authors perform an extensive evaluation of several MIA approaches applied to LLMs, on several datasets and models. I think that this benchmarking of methods is valuable for the community. The authors also make their code available.

- The evaluation reveals the intringuing finding that current MIA perform quite poorly. As this finding directly challenges results from some of the previous works, and a plausible explanation is offerred, I believe that publishing the paper can be beneficial to the community.

- An important finding is that the MIA decision boundary is not robust to small changes in the text. The authors show that a slightly modified version of a member text that would be considered as a member by a human (but is technically a non-member as per the strict MIA definition) would be classified as a non-member.  This challenges the current definition of MIAs and could motivate interesting developments in the field.

**Reasons To Reject:**

- Potential methodological flaw: the Reference-based and min-k prob attacks seem to be tuned on the test data. I re-read Appendices A.3.1. and A.4 several times looking for a mention of validation data and I found only one, for the DATABLATIONS model, while the Reference-based tuning is done on PYTHIA-DEDUP models (for which there is no mention of validation data). Tuning the methods on the same data on which final performances are reported is a methodological flaw because it biases the results. It also gives an unfair advantage to the methods being tuned compared the ones which are not tuned. This is especially problematic here since the extensively tuned Reference-based attack is found to perform « best » in the end. This can be fixed by using a validation dataset for tuning the methods which is independent from the test set.

- Unclear exactly how the author’s analysis of temporal shift relates to the previous works of Shi et al, 2023 and Meeus et al, 2023. The paper should clarify whether the same experiment settings (LLMs, data domains, temporal ranges) as the two papers are reproduced, or if the author only speculate that these two papers suffer from temporal shift. Making this very precise is needed to support the abstract claim that « this apparent success can be attributed to a distribution shift », else the claim should be more nuanced. For instance, going through these two papers, I understand that in the arXiv domain, Shi et al uses members and non-members from before 2016 and after 2023, respectively, Meeus et al uses members before February 2023 and non-members after February 2023, and the current paper uses members from July 2020 and non-members from later months. Since the temporal shift is slow (Figure 6), it likely affects the former paper more than the latter. Furthermore, Meeus et al. also uses a second dataset where there is no temporal shift and uses longer documents for which MIA risk is likely to be higher. These differences are not discussed in the paper.

- Limited evaluation: the paper focuses on 200-word texts. When it comes to copyright infringment, longer documents such as news articles and books pose a bigger concern. As mentioned by the authors and shown by Shi et al, 2023, MIA risk increases with text length. Therefore, it is likely that MIAs might still work for, e.g., books. If this is true, the paper’s core claim would not hold. There is no discussion of this in the paper.

- A point is made that the MIA definition is not aligned with how we reason about privacy, but no better definition is proposed. The link between this part and the rest of the paper is not very clear. The usefulness of experimenting with large edit distances of 10 and 25 is unclear to me: if a large fraction of the text is replaced with random tokens, would the text still be considered to be a member by a human? Even for an edit distance of 1, replacing a token with a random token does not guarantee that the text still makes sense (which would be necessary in any notion of neighborhood). The empirical part of this section seems insufficiently developed.

---

> ### Author Rebuttal · Authors · 2024-05-31
>
> We thank the reviewer for their helpful feedback and for supporting our paper. We respond to the reviewer’s main concerns below, and will address questions/comments in a follow-up post:
>
> *[Reason to reject 1]* No details on hyperparameter tuning
>
> We clarify that none of the evaluated attacks involve any tuning. Reference models are open-source, out-of-the-box pretrained LMs and the choices for k are those suggested in Shi et al.
>
> Regarding “best” hyperparameter choice, we aim to show that even in the best case, MIA scores are significantly lower than what would be expected based on trends in the MI literature.
>
> *[RR2]*  Temporal shift setting is inconsistent to Shi et al & Meeus et al.
>
> We will clarify that we analyze temporal shift to show how unintentional distribution shifts may arise in an otherwise practical selection of non-members for pretrained LLMs, not to claim that the results in these papers are solely due to such shifts.
>
> We don’t use the exact setups from prior work as (1) target LMs like LLaMa don’t release training data needed to assess n-gram overlap or (2) their non-members are not publicly released and are difficult to replicate.
>
> *[RR3]* Evaluation limited to 200-word text
>
> Shi et al, 2023 evaluates texts up to 256 words, which is close to the text length we use. While an increase in performance with length is possible, that experiment is orthogonal to ours; inherent differences in LLM training and MI evaluation (i.e., benchmark candidates, membership definition) would still impact evaluation on longer texts.
>
> *[RR4]* No better definition proposed
>
> The discussion of high n-gram overlap of non-members with members naturally raises the question of whether exact membership is always useful concerning information leakage. We propose approximate membership to motivate aligning membership definitions with respective privacy use cases [1] (i.e., partial lexical match, semantic leakage) rather than an exact new definition of membership. We will clarify this.
>
> Random token replacement is naive. However, it’s a first step in exploring approximate membership definitions; intact portions of modified members can still leak member information. We also explore a more realistic definition in semantic neighbors. We aim to demonstrate that current MIAs can be too sensitive to capture what humans perceive as “close” to members and encourage future work to build on this in parallel to standard MI.
>
> ### References
>
> [1] https://arxiv.org/abs/2202.05520

---

> > ### Author Response · Authors · 2024-05-31
> >
> > We follow up on questions and comments below and will address the minor comments in our updated revision
> >
> > [Q1] The validation/test data of the Pile is sampled uniformly at random from the training data ([1], Section 3.1)
> >
> > [Q2] We appreciate the reviewer bringing these papers to our attention and will include discussion of earlier notions of membership neighborhoods. However, these works seem target a different direction, focusing on generating non-members such that they are falsely identified as members. On the other hand, we explore how neighborhood non-members can still result in information leakage of members, but current MIAs can be too sensitive to classify this member information leakage.
> >
> > [Q3] We choose to only evaluate blackbox MIAs without stronger data access assumptions for this paper.
> >
> > [Q4] We use 1000 bootstrap samples to create the C.I. (See Appendix A.4)
> >
> > [Q5] We plotted the MIA scores assigned to non-members against their respective {7,13}-gram overlaps; we will add these figures to the Appendix. However, as mentioned in the response to Q3 for reviewer F61L, the important takeaway to clarify in the future revision is that non-member distribution shift (i.e., in n-gram overlap) can inflate MIA, which we felt was better demonstrated by the current design.
> >
> > [Q6] While we did not include exact statistical significance tests (will be included in revision), The error bars of our results are small enough for our results to imply statistical significance. We will include numbers for these statistical significance tests.
> >
> > [Q7] We spot checked that documents in the Pile when passed through our pipeline with the ArXiv API from the raw source yielded the same output. We don’t use ArXiv S3 due to paywall.
> >
> > [Comment 1] Agreed
> >
> > [C2] We agree it is more similar to reference calibration [2] and will revise accordingly.
> >
> > [C3] We chose to keep this analysis in the appendix due to space limitations and because we felt a larger ablation over reference model choice was needed to substantiate the plausible analysis.
> >
> > ### References
> >
> > [1] https://arxiv.org/pdf/2101.00027
> >
> > [2] https://arxiv.org/abs/2111.08440

---

> > > ### Comment · Reviewer_sAnK · 2024-06-03
> > > **Response to authors**
> > >
> > > Thank you for the response. I have changed my score following the authors' response, as I believe that the reasons to accept the paper outweigh the limitations. However, I still think that the paper should be revised to include the discussion points of the authors' rebuttal and the limitations, as outlined below.
> > >
> > > [RR1] Thanks for clarifying this. The authors experiment with multiple reference models and pick the best one for all the remaining analyses, which are performed on the same test set. This can be regarded as a form of tuning and could bias the results. However, looking at subsequent plots, the results and trends identified look stable, which makes me think that the impact of this is quite limited.
> > >
> > > [RR2], [RR3] and [RR4] Thanks for the clarifications, please consider adding the information and context provided in your response to the relevant section. Regarding [RR3], I still think that evaluating on 200-word texts limits the scope of the claim (and title) that MIAs in general do not work against LLMs; if the evaluation is limited to short texts then the paper mainly challenges current evaluations on short texts like Shi et al., 2023. Please include a discussion of this limitation and how it possibly impacts the main claim.
> > >
> > > [Q1] Can the authors clarify what they mean by validation and test data being sampled from the training data? Does the training data overlap with validation and test data?
> > >
> > > [Q2] Thanks for the clarifications.
> > >
> > > [Q3] Thanks for clarifying this very important point. Please consider acknowledging this limitation: as only score-based attacks are benchmarked, it remains possible that results of MIAs can be improved if an ML classifier was trained for the task (as opposed to score-based MIAs). The authors construct a test set of in-distribution members and non-members and it isn't clear why a similar training set cannot be constructed.
> > >
> > > [Q4]-[Q7] and [C1]-[C3] Thanks for the clarifications.

---

> > > > ### Author Response · Authors · 2024-06-07
> > > > **Response to reviewer**
> > > >
> > > > We thank the reviewer for the helpful discussion. We agree that the discussion points above would be valuable in the future revision and will add them alongside the minor comments from before. Some additional notes:
> > > >
> > > > [RR1] To further clarify, we bolster the observation that MIAs perform near-random when targeting pre-trained LLMs (from the main experiments) by showing the empirically determined best performance of the MIAs. Here, the hyperparameter ablations for reference model or k choice is to form a rough empirical "upper-bound" for these attacks' performances. Evaluating MIAs for the sake of determining the best-performing MIA would indeed be biased if such ablations for hyperparameters were done on the same evaluation benchmark, but this is a different goal from what we aim to show in our main experiments.
> > > >
> > > > For the other analysis, we saw that such hyperparameter choices didn't impact the observed trends and chose not to visualize the figures for other hyperparameter choices due to space limitations.
> > > >
> > > > [RR2-4] We will add these clarifications and limitation discussions in the next revision
> > > >
> > > > [Q1] Per the Pile paper, the validation and test sets are sampled randomly from the complete Pile and then held out from the training data. Any overlap between the validation/test sets and the training data is due to the nature of the complete data (as discussed in Section 3.2.2 and in the Pile paper itself), not the sampling/splitting.
> > > >
> > > > [Q3] Using a classifier could yield greater performances. However, this threat model is different as it requires access to a ground truth set of members/non-members, even if only a small set is needed. We will clarify this difference in the future revision.

---

### Official Review · Reviewer_F61L · 2024-05-06

**Rating:** 6
**Confidence:** 4
**Ethics Flag:** 1

**Summary:**

The paper evaluates 5 recent membership inference attacks (MIAs) for LLMs and finds that most of them have near-random (AUC ROC < 0.6) performance. The evaluation was carried out on the Pythia suite of models trained on the PILE dataset. This setup provides an appropriate test bed since the knowledge of the train and test sets enables the fair evaluation of MIAs. It is suggested that the following characteristics of LLMs substantially reduce the performance of MIAs: (1) use of massive training data, (2) near-one epoch training for LLMs, (3) frequent overlap of members and non-members in natural language (non-members have high n-gram overlap with members).

It is suggested that the high performance of the previous MIA (Shi et al., 2023) was due to the temporal shift.

**Questions To Authors:**

- Did the authors consider other metrics to analyze the overlap between members and non-members apart from n-grams, for example, MAUVE score: https://arxiv.org/pdf/2102.01454? If so, which kind of other metrics are relevant?
- Why the smaller models from Pythia were not used in the assessment (the model of size < 160 min parameters)?
- It is claimed that “MIAs perform significantly better when the non-member distribution has a lower n-gram overlap“. Then, why does the GitHub dataset have a much higher MIA than other datasets (in Table 2, Orig for GitHub vs other datasets like Wikipedia)?

**Reasons To Accept:**

- The usage of the Pythia suite of models and the PILE dataset is very good choice for the task of assessing the performance of MIAs on LLMs. Overall, the experiments are thorough and leverage the underlying Pythia suite of models. For example, I appreciate that the authors employ the PYTHIA-DEDUP model suite’s intermediate checkpoints to assess the impact of different amounts of training data.
- The provided analysis of the overlaps between members and non-members using the n-grams is relevant, though might not be precise enough.

**Reasons To Reject:**

- The paper states negative results for the MIA attacks which makes the practical assessment of privacy leakage from LLMs more questionable. No alternative solutions are proposed.

---

> ### Author Rebuttal · Authors · 2024-05-31
>
> We thank the reviewer for their helpful feedback and for supporting our paper.
>
> *[Reasons to Reject]* Lack of alternative solutions for practical assessment of privacy leakage from LLMs
>
> Section 5 proposes the alternate notion of approximate membership (and explores two concrete definitions: edit-based and semantic-based neighbors) to align the goals of MI with privacy settings, where auditors/adversaries can tune the approximate membership definition to fit their privacy use case [1]. We believe this is a promising direction to explore in future work.
>
> *[Q1]* Other overlap metrics
>
> We chose n-gram overlap as it is an established method in decontamination [2] and is fast to compute. We also consider additional metrics, e.g., closest semantic embedding similarity, another decontamination metric [3]. We can also use MAUVE scores; however, these methods require generating embeddings, which with a large-enough model is computationally expensive, so we opt for the simpler and cost-effective n-gram overlap.
>
> *[Q2]* Smaller models from Pythia (< 160M)
>
> We didn’t include them initially as our target was large LMs, but we will include it in the next revision. Results of main experiments on Pythia 70M-deduped on select domains:
>
> ArXiv -
> LOSS: .494, Ref: .519, Min-k: .501, Zlib: .505, Ne: .504
>
> HackerNews -
> LOSS: .506, Ref: .505, Min-k: .493, Zlib: .503, Ne: .494
>
> Wikipedia -
> LOSS: .497, Ref: .496, Min-k: .506, Zlib: .492, Ne: .490
>
> *[Q3]* GitHub as an outlier
>
> We understand that this claim is currently insufficient. Instead, it would be more accurate to say “MIAs perform significantly better as the non-member distribution concentrates towards lower n-gram overlaps, diverging away from the natural n-gram overlap distribution of held out datapoints from the training data (against the rest of the training data)”.
>
> For the case of GitHub, we refer to Appendix B.3. In Fig 7, we see the soft decontamination (Orig) already yields a significant distribution shift for the GitHub non-members. In other domains, a 7-gram overlap threshold of <= 20% is not as significant of a distribution shift relative to their natural 7-gram overlap distribution (Fig 14); MIA results over the Orig GitHub dataset are thus similar to those on the artificially thresholded benchmarks in other domains. We will update and clarify this claim accordingly in the revision.
>
> ### References
>
> [1] https://arxiv.org/abs/2202.05520
>
> [2] https://arxiv.org/abs/2005.14165
>
> [3] https://arxiv.org/pdf/2311.04850

---

> > ### Comment · Reviewer_F61L · 2024-06-05
> > **Thank you**
> >
> > Thank you for your answer, I appreciate that. I keep my positive score.

---

### Official Review · Reviewer_NqZv · 2024-05-10

**Rating:** 8
**Confidence:** 3
**Ethics Flag:** 1

**Summary:**

Summary:
The paper studies Membership Inference Attacks on LLMs. The authors perform many analyses to study the successes and failures of MIA attacks in the specific case of LLMs. In particular, the authors found that MIA performs surprisingly badly for LLMs. The paper provides an extensive suite of experiments to test various hypotheses to explain what makes MIA so hard for LLM. The paper concludes with some remarks about reframing the task of Membership Inference for the LLM scenario.


Overall:
I think this is a good paper worthy of publication.

**Reasons To Accept:**

This is a well-executed paper. It measures behavior with a careful and meaningful experimental setup, proposes several hypotheses to explain the observations, and crafts controlled experiments to verify the hypotheses. The paper also positions itself with respect to the bigger field of MI and makes some recommendations for better definitions in the future. Overall, it was an insightful paper.

**Reasons To Reject:**

The claim that MIA performs near random could downplayed a bit because Fig 1. seems to show performance statistically significantly above chance in most scenarios. To make the claim that MIA performs badly for LLM, it could be more convincing to compare the reported AUC with AUCs in order domains where MIAs have been developed and tested before.

---

> ### Author Rebuttal · Authors · 2024-05-31
>
> We thank the reviewer for their helpful feedback and for supporting our paper.
>
> *[Reason to Reject]* Compare AUCs of MIAs in other domains MIAs have been established in.
>
> Thank you for the suggestion. We will add relevant references (and numbers) for other domains such as tabular [1, 2], image [3, 4], and text [5] data, along with results for LLMs with fine-tuning data [6].
>
> ### References
>
> [1] https://arxiv.org/abs/1709.01604
>
> [2] https://arxiv.org/abs/1908.11229
>
> [3] https://arxiv.org/abs/2112.03570
>
> [4] https://arxiv.org/abs/2111.09679
>
> [5] https://arxiv.org/abs/2203.03929
>
> [6] https://arxiv.org/abs/2311.06062

---

### Decision · Program_Chairs · 2024-07-10

**Decision:**

Accept

**Comment:**

This is an empirical paper which evaluates the effectiveness of membership inference attacks against LLMs, concludes that they are ineffective most of the time, and goes on to investigate the factors that make the attacks more or less effective. It especially investigates the effect of near-online training and the inherent ambiguity in what counts as membership.

Overall, this is a solid empirical paper. It first demonstrates an interesting central claim, and then analyzes it from multiple angles. The topic is very timely with the increasing concern over copyright/plagiarism. The writing is clear, and the experiments seem well motivated. Reviewers raise a few concerns with the details of the experiments, such as the tuning methodology and possible inconsistency with prior work. But at the end of the day, all of the reviewers feel this paper deserves to be accepted, and this is my sense as well.

[At least one review was discounted during the decision process due to quality]